# LECTOR: Joint Optimization of Scientific Reasoning Graphs and Introduction Generation

**Jiabei Xiao** [1 2]  **Yizhou Wang** [1 2]  **Chen Tang** [1 2]  **Pengze Li** [2 3]  **Wanli Ouyang** [1 2]  **Shixiang Tang** [1]

## Abstract

AI Scientists have shown promising progress across multiple stages of the research pipeline, among which automatic scientific paper writing remains a formidable challenge. The Introduction writing is especially challenging, which demands not only linguistic fluency, but logical soundness and verifiable faithfulness. Most AI-assisted methods treat the task as text generation instead of reasoning and structuring, leading to severe drawbacks, *e.g.*, hallucinating citations. To address this, we first formulate the Content-Conditional Introduction Generation (CCIG) task, which requires grounding the Introduction in the paper's core evidence. We then propose LECTOR, a novel Logic-Expression Co-Reinforcement Learning framework that can strictly follow the scientist's logic, add high-quality citations and keep structured expressions. LECTOR first constructs a logic-reasoning graph from the paper's main body to serve as a verifiable logical blueprint. Subsequently, it employs a Logic-Expression Co-Rewarding mechanism to jointly optimize for both the graph's structural fidelity and the final narrative's quality. We conduct a dataset from Nature Communications papers to assess our method. Extensive experiments show consistent improvements in both logic fidelity and Introduction generation quality metrics, *e.g.*, Graph Quality (**+26.7%**), Citation Quality (**+8.6%**), and Paper Consistency (**+3.3%**). Code and data are available at https://github.com/Xiao-Youth/LECTOR.

---

[1]The Chinese University of Hong Kong [2]Shanghai Artificial Intelligence Laboratory [3]Fudan University. Correspondence to: Shixiang Tang <tangshixiang2016@gmail.com>.

*Proceedings of the 43$^{rd}$ International Conference on Machine Learning*, Seoul, South Korea. PMLR 306, 2026. Copyright 2026 by the author(s).

## 1. Introduction

Recent advances in large language models have enabled the development of *AI Scientist* systems, which aim to automate the scientific research process. This ambition is exemplified by the development of massive, closed-source models like OpenAI's Prism, which target scientific writing (OpenAI, 2026). While such models may achieve high linguistic fluency, their "black-box" nature renders their internal reasoning processes unverifiable. In scientific writing, the Introduction section is particularly critical in summarizing the entire research: a high-quality Introduction requires not only fluent language generation but also accurate understanding of research logic and structured presentation of motivation, methodology, and contributions. Consequently, *an AI's capacity to generate a logically sound and well-structured Introduction serves as a critical benchmark*, distinguishing deep comprehension from superficial text generation rather than merely producing surface-level text.

Yet, existing AI-assisted writing methods always fail to meet this benchmark. The core reason is that they treat *Introduction* writing as a common text generation problem, when it is fundamentally a task of reasoning and structuring. It requires abstracting the paper-level reasoning structure from technical content and only then transforming it into a coherent high-level narrative. Most methods simply design writing prompts for general LLMs, a black-box approach that bypasses the crucial reasoning step entirely, leading to severe drawbacks that compromise academic integrity. First, these methods can result in hallucinating citations, *e.g.,* nonexistent publications or incorrect authorship. Second, and more critically, they fail to ensure logical consistency between the *Introduction* and the following *Results* and *Methodology*. As a result, generated Introductions often exhibit logical inconsistencies, missing motivations, or misaligned contributions.

To systematically address these failures, we propose a new content-conditioned introduction generation (CCIG) task, a more serious one for introduction generation, that we ask the model to write the *Introduction* section given the *Methodology*, *Results*, *Analyses*, and the *Citation* list. To seriously evaluate the logic, citation and expression of the generated introduction, we design a set of metrics including logic fi-

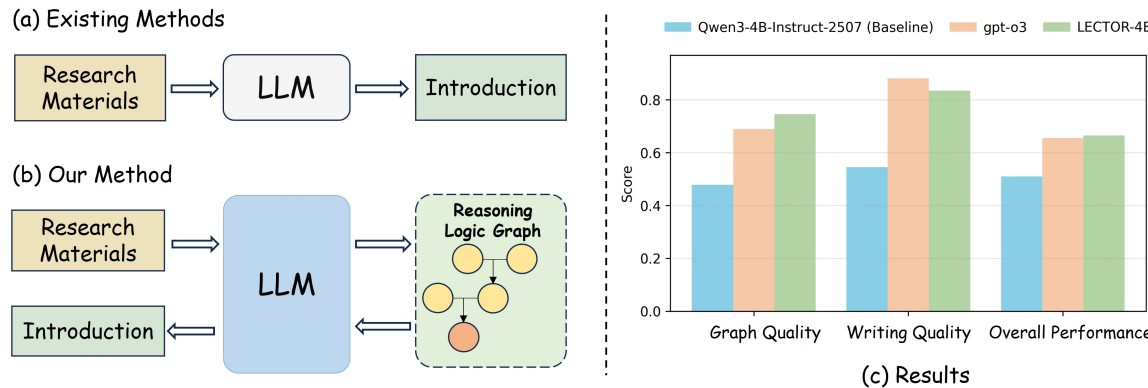

Figure 1. **Overview of the LECTOR framework and performance evaluation.** **(a) Existing methods** treat introduction writing as a direct text generation task, often leading to logical inconsistencies and hallucinations. **(b) LECTOR** reformulates the task as *Content-Conditional Introduction Generation* (CCIG), first extracting a **Reasoning Logic Graph** as a verifiable logical blueprint to guide logic-aware writing. **(c) Results** show that our **LECTOR-4B** significantly outperforms the Qwen3-4B baseline. Notably, LECTOR-4B achieves superior **Overall Performance** compared to the state-of-the-art commercial closed-source model **GPT-o3**, validating the effectiveness of our logic-expression co-reinforcement learning approach.

delity, expression fluency, and citation quality. Together, the task and our metrics provide a principled framework for developing and benchmarking models that not only generate fluent but also logically self-contained introductions.

To solve the CCIG task, we introduce **LECTOR**[1], a Logic-Expression Co-Reinforcement Learning framework. The key innovations are two-fold. First, we leverage a logic-reasoning graph as a structured intermediate representation to regularize the logic of the generated introduction. In the logic-reasoning graph, nodes are self-contained sentences that represent information from the paper, while edges explicitly model the logical relationships that connect these claims into a coherent argument, guided by the three Peircean reasoning paradigms (Peirce, 1992), The graph acts as an explicit logical blueprint, forcing the model to first map out the paper's argumentative skeleton before generating any text. Second, we propose a logic and expression co-rewarding, where reward signals are computed from both the quality of the extracted reasoning structure and the generated Introduction, encouraging the model to align logic fidelity with writing quality. This joint optimization strategy enables mutual reinforcement between scientific understanding and structure-aware writing.

To validate the effectiveness of the proposed method, we construct a large-scale dataset of 10,200 scientific papers from *Nature Communications*, covering diverse physics-related domains and spanning publications from April 2010 to March 2025. Using this dataset, we evaluate our approach on the challenging task of logic-aware Introduction writing.

LECTOR allows LLMs to bridge the gap between deep logical reasoning and high-quality narrative generation. To demonstrate this, we implement LECTOR on a 4B-parameter model, *i.e.*, Qwen3-4B-Instruct-2507 (Yang et al., 2025), observing remarkable improvements across all metrics, including Graph Quality (**+26.7%**), Citation Quality (**+8.6%**), and Paper Consistency (**+3.3%**). Notably, the final performance of our lightweight model is comparable to that of strong commercial systems like GPT-o3 (OpenAI, 2025), while vastly outperforming its untrained baseline. This demonstrates that by explicitly modeling a paper's reasoning structure and jointly optimizing for logic and expression, our framework provides a more efficient path to high-fidelity scientific writing than relying on model scale alone.

Our contributions are summarized as follows: (1) We introduce the content-conditional introduction generation (CCIG) task, a new and more rigorous task for scientific writing AI, which prioritizes verifiable logical fidelity over mere topical fluency. (2) We propose LECTOR, a novel Logic-Expression Co-Reinforcement Learning framework designed to solve the CCIG task, which utilizes a *logic-reasoning graph* as an explicit intermediate representation and a *co-rewarding mechanism* to jointly optimize for both structural logic and narrative quality. (3) We construct a dataset using papers from Nature Communications and empirically validate that LECTOR improves both logic fidelity and Introduction generation quality.

## 2. Related Work

### 2.1. LLM-based Scientific Writing

Recent advances in Large Language Models (LLMs) show the potential for automating scientific writing. While rais-

---

[1]Lector in Latin means Reader in English, reflecting the model's goal of deep comprehension during writing.

ing lots of concerns about academic misconduct (Cheng & Zhang, 2025; Kwon, 2025), top-tier venues such as ICML, Nature and Science open a window to AI-assisted paper writing with rigorous regulations. One common use case is to generate literature surveys. These methods generally leverage LLMs to automatically collect relevant papers and synthesize coherent survey articles, demonstrating the potential of LLMs in large-scale academic content generation (Wang et al., 2024b; Yan et al., 2025; Zhang et al., 2025). Beyond survey writing, recent AI Scientist systems (Lu et al., 2024; Yamada et al., 2025; Weng et al., 2025b; Yu et al., 2025; Tang et al., 2026; Weng et al., 2025a) further extend this direction by generating complete academic papers in an end-to-end manner. Despite the impressive progress in end-to-end paper generation, these systems often suffer from quality issues in writing, including logical inconsistency, unclear contribution positioning, and weak structural organization (Ivanov, 2025; Mezzadri, 2025). This indicates that directly generating papers without explicitly modeling research logic may limit the reliability and interpretability of the writing process (BaHammam, 2025; Knöchel et al., 2025), motivating a deeper investigation into how scientific writing should be guided by structured understanding. The most relevant work is SciIG (Garg et al., 2025), which systematically benchmarks LLMs on the task of writing research paper Introductions, providing detailed evaluation of writing quality across different models (Liu et al., 2024; Team et al., 2025). While this work offers valuable insights into the strengths and limitations of current LLMs in scientific writing, it primarily leverages titles, abstracts to generate introductions and therefore focuses on expression fluency and does not explicitly evaluate whether the generated text is grounded in a correct understanding of the underlying research logic. In contrast, our work leverages a reasoning-logic graph to regularize the flow of introduction so that it follows the logic of *Results, Methodology, Analysis and Citations* and designs a logic-expression co-rewarding strategy to improve both logic and expression of the generated introduction.

## 2.2. Structured Representation for Scientific Documents

Structured representations of scientific documents extract the internal reasoning-logic in scientific papers, which show benefits to a variety of downstream understanding tasks. Open Research Knowledge Graph (ORKG) (Jaradeh et al., 2019) represents research contributions as semantic entities and relations to support scholarly comparison and retrieval. NLP-AKG constructs a large-scale academic knowledge graph for NLP by extracting fine-grained conceptual relations across papers, enabling structured semantic search and analysis (Lan et al., 2025). Contrastive Hierarchical Discourse Graph models scientific papers with hierarchical discourse structures for summarization (Zhang et al., 2023).

These approaches demonstrate the effectiveness of structured representations for downstream tasks such as retrieval and summarization. Recently, ARCHE (Li et al., 2026) introduces a benchmark for extracting latent reasoning chains from scientific papers, explicitly targeting the recovery of implicit reasoning structures and revealing the limitations of current LLMs in capturing formal reasoning processes. However, it focuses on reasoning extraction alone and does not connect structured reasoning representations to scientific writing. In contrast, our work leverages a reasoning logic graph to explicitly model research-level logical structure and directly uses it to guide Introduction generation.

## 2.3. Benchmarks for Paper Understanding

Early benchmarks mainly evaluate information-seeking question answering over scientific documents. PubMedQA focuses on biomedical literature QA (Jin et al., 2019), while QASPER extends this setting to expert-authored questions requiring multi-section evidence aggregation (Dasigi et al., 2021). Recent datasets target deeper document-level comprehension. SciDQA emphasizes cross-section reasoning for scientific reading comprehension (Singh et al., 2024). With the emergence of long-context models, Long-Bench and LongBench v2 benchmark LLMs on realistic long-document understanding tasks, including scientific papers (Bai et al., 2024; 2025). However, existing benchmarks primarily assess local comprehension, retrieval, or long-context reading. In contrast, our work evaluates research-level understanding by explicitly modeling scientific reasoning logic and assessing it through structure-guided Introduction generation.

## 3. Methodology

While Large Language Models (LLMs) can generate fluent and plausible scientific introductions, their outputs often lack deep logical coherence and verifiable fidelity to the core research narrative. This limitation arises because conventional training paradigms optimize for textual coherence on surface-level textual patterns, rather than explicitly modeling the underlying logic graph of scientific arguments. Existing introduction generation tasks ask the LLM to write the introduction based on the title, abstract and citations, which only contains more abstract information. We consider such a setting to contradict the real academic writing scenario, where the introduction section is summarized from more detailed parts such as results, methodology, analysis and citations. Therefore we propose a Content-Conditional Introduction Generation (CCIG) task (Sec. 3.1) and teach an LLM to generate a coherent and logically faithful Introduction for scientific papers given results, methodology, analysis sections and citation lists. To build a simple baseline, we propose **LECTOR**, a Logic-Expression

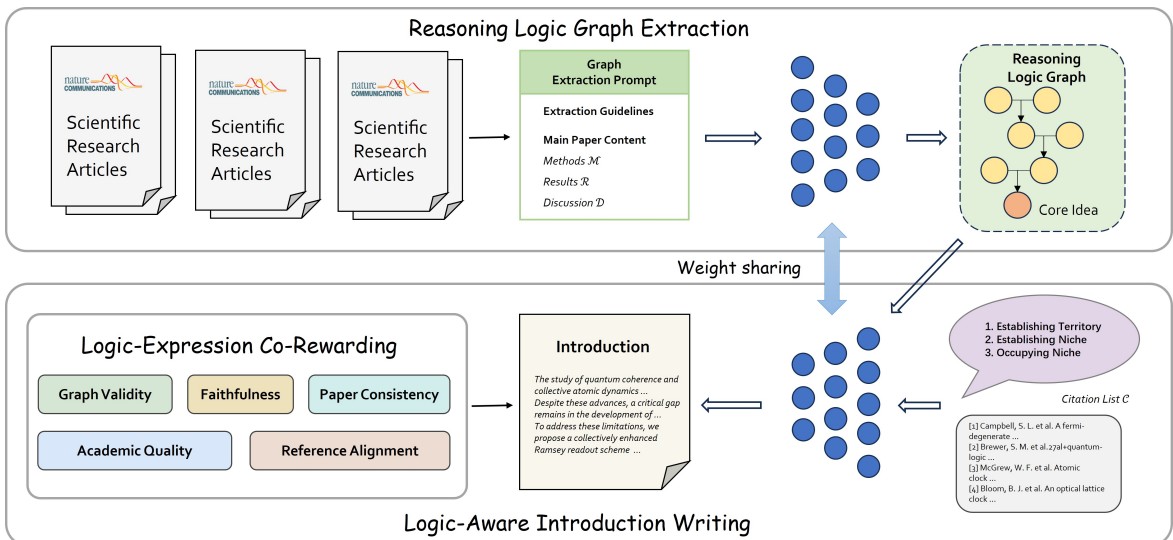

*Figure 2.* **The overall architecture of LECTOR.** The framework operates in two synergistic stages within a single rollout: **(Top) Reasoning Logic Graph Extraction:** Given the main body of scientific research articles including Methods ($\mathcal{M}$), Results ($\mathcal{R}$), and Discussion ($\mathcal{D}$) but excluding the Introduction, LECTOR extracts an explicit **Reasoning Logic Graph**. This graph consists of nodes connected through *deduction*, *abduction*, and *induction* to derive the paper's core idea. **(Bottom) Logic-Aware Introduction Writing:** Taking the extracted graph and a citation list $\mathcal{C}$ as input sources, the model generates a structured introduction following the **CARS (Create a Research Space)** move structures (e.g., *Establishing a Territory/Niche*). **Optimization:** Both stages share weights and are jointly optimized through a **Logic-Expression Co-Rewarding** mechanism. By rigorously evaluating *Graph Quality* and *Graph-Writing Alignment* alongside *Writing Quality* and *Citation Quality*, LECTOR ensures that the high-quality reasoning logic graph effectively grounds the final introduction to be logically sound, verifiably faithful, and narratively fluent.

Co-Reinforcement Learning framework, where a logic reasoning graph (Sec. 3.2) behaves as a versatile intermediate representation to enhance the logic of generated introduction, and a logic and expression co-rewarding (Sec. 3.3) designed to jointly optimize for logic fidelity and narrative quality.

### 3.1. Content-Conditional Introduction Generation Task

Different from existing formulation that leverages title, abstract, citation lists to generate introduction, we propose a more realistic setting that is to generate the introduction section based on more detailed experimental information. Specifically, given a scientific paper $\mathcal{P}$, we define its main body $\mathcal{B}$ as the content excluding the Introduction $\mathcal{I}$. The main body, which comprises the **Methods** $\mathcal{M}$, **Results** $\mathcal{R}$, **Analyses** $\mathcal{A}$, and **Citations** $\mathcal{C}$, serves as the detailed, low-level evidence that substantiates the claims of the paper. The content-conditional introduction generation (CCIG) task requires the model to do a mapping from this evidence to the high-level narrative of the Introduction:

$$\mathcal{I} = \mathcal{F}(\mathcal{M}, \mathcal{R}, \mathcal{A}, \mathcal{C}), \tag{1}$$

where $\mathcal{I}$ is the introduction section of the paper, $\mathcal{M}$ is the method of the paper, $\mathcal{R}$ is the result section of the paper, $\mathcal{A}$ is the analysis of the paper and $\mathcal{C}$ is the citation of the paper.

Our task formulation differs fundamentally from concurrent work like SciIG (Garg et al., 2025), which generates an In-

troduction conditioned on high-level summaries (*e.g.*, Title, Abstract) and external context (*e.g.*, Related Papers). While SciIG's setup prioritizes topical relevance and summary, our CCIG task focuses on **logic grounding** and **writing quality**. By conditioning on detailed information in the main body, *e.g.,* method, results, analysis and citation lists, we create a more realistic scenario that requires a model to ground its narrative in the specific methods and findings of the research paper, rather than summarizing high-level concepts.

### 3.2. Logic-reasoning Graph as an Intermediate Representation

The direct and end-to-end approach (*i.e.*, LLM supervised finetuning) to content-conditional introduction generation is sophisticated and ill-posed. This approach forces a model to simultaneously comprehend a long, unstructured body of text and compose a logically coherent narrative, leading to two critical problems. First, it lacks guidance for the LLM to focus on the pivotal elements of the research. Second, it lacks an explicit mechanism to enforce logical consistency.

To overcome these issues, we argue that an intermediate representation that regularize the writing logic is not only beneficial, but essential. An effective representation must possess two properties: (1) **Compactness**, which helps distinguish core arguments from the verbose details of the main body, and (2) **Structuring**, which helps synthesize concepts and articulate the logical connections among them.

*Table 1.* Definitions of the Six Reasoning Edge Types.

| Paradigm | Edge Type | Role: Represents a premise that is... |
|---|---|---|
| Deduction | `deduction-rule` | A general principle, law, or established rule. |
| | `deduction-case` | A specific instance or case that falls under that general rule. |
| Induction | `induction-common` | A general pattern or commonality abstracted across multiple observations. |
| | `induction-case` | An individual observation or piece of evidence supporting the pattern. |
| Abduction | `abduction-phenomenon` | An observation or phenomenon that requires an explanation. |
| | `abduction-knowledge` | Background knowledge that offers the best explanation for the phenomenon. |

Therefore, we consider that the **logic-reasoning graph** is the ideal intermediate representation for scientific introduction writing. Dissimilar to unstructured summaries, a graph explicitly models the foundational components of scientific reasoning. By converting the unstructured main body $\mathcal{B}$ into a structured logic graph $\mathcal{G}$, we transform the task from a complex, implicit reasoning problem into a more tractable, logic-guided generation problem.

### 3.2.1. REASONING LOGIC GRAPH EXTRACTION

From the main body $\mathcal{B} = (\mathcal{M}, \mathcal{R}, \mathcal{A}, \mathcal{C})$, the model constructs a reasoning logic graph $\mathcal{G}$, which serves as an explicit, formalized representation of the relationships among research problems, methods, experiments, and findings:

$$\mathcal{G} = \mathcal{F}(\mathcal{B}) = \mathcal{F}(\mathcal{M}, \mathcal{R}, \mathcal{A}), \qquad (2)$$

where $\mathcal{G}$ is the logic-reasoning graph defined below. $\mathcal{F}$ is initialized from Qwen-4B, prompted with descriptions of the Reasoning-logic graph $\mathcal{G}$, and finetuned by reinforcement learning with Logic-Expression Co-rewarding (Sec. 3.3). To force the model to concentrate on the underlying logic of the paper, we omit bibliographic details, *i.e.,* Citation $\mathcal{C}$.

**Definition of Reasoning Logic Graph $\mathcal{G}$.** Our graph formalism is inspired by the philosophical work of Charles S. Peirce (Peirce, 1992), who categorized all valid reasoning as *deductive*, *inductive*, or *abductive*, or combinations thereof. A reasoning Logic Graph $\mathcal{G} = (\mathcal{V}, \mathcal{E})$ is a single-rooted directed acyclic graph, where its nodes $\mathcal{V}$ are complete, self-contained sentences that represent an atomic unit of information extracted from the paper, *i.e.*, a scientific claim, an experimental finding, a piece of background knowledge, an opinion or statement derived from referenced work. The edges of the graph $\mathcal{E}$ are designed to model the three Peircean reasoning paradigms. Each logical inference is formed by a specific pair of premise edges pointing to a single conclusion, ensuring that every reasoning step is well-founded and traceable. The roles of the six defined edge types are detailed in Table 1.

**Discussion.** While our reasoning graph shares a motivational origin with the one used in the ARCHE benchmark (Li et al., 2026), their purposes are fundamentally different.

ARCHE employs its graph as a final output for the *evaluation* of an LLM's reasoning capabilities. In contrast, we utilize the reasoning graph as an *intermediate representation* designed to guide the learning of content-conditional introduction generation.

### 3.2.2. LOGIC-AWARE INTRODUCTION WRITING

To produce a coherent and academically polished introduction, we borrow the guidelines from the influential **CARS (Create A Research Space) framework** (Swales, 1990). This framework posits that a good introduction consists of three moves: (1) The first move is to establish the territory, which is describing the broader research area and its importance, summarizing the relevant background knowledge represented in the logic-reasoning graph $\mathcal{G}$; (2) The second move is to build the niche, which is identifying gaps, unresolved issues, or limitations suggested by the logic reasoning graph $\mathcal{G}$; (3) The last move is to present the central research idea corresponding to the root node of the graph, showing how it logically follows from the preceding reasoning steps. Based on the above guidelines as prompts, we leverage a learnable large language model $\mathcal{F}$ to generate the introduction $\mathcal{I}$, *i.e.,*

$$\mathcal{I} = \mathcal{F}(\mathcal{G}, \mathcal{C}), \qquad (3)$$

where $\mathcal{I}$ is the generated introduction, $\mathcal{C}$ is the citation list, and $\mathcal{F}$ is the large language model that can be trained by the reinforcement learning in Sec. 3.3. The citation list $\mathcal{C}$ provides all bibliographic entries, each associated with a unique index. The model is constrained to rely exclusively on $\mathcal{G}$ for the scientific narrative and on $\mathcal{C}$ for sourcing citations, which must be inserted using the required `[idx]` format.

### 3.3. Logic-Expression Co-Reinforcement Learning

While the task can naturally decompose into two stages *(i)* Reasoning Logic Graph Extraction and *(ii)* Logic-aware Introduction Writing, training these components in isolation poses significant challenges. A disjoint two-stage training paradigm not only requires expensive annotation for intermediate graph representations (logic graphs aligned with spe-

cific introductions) but also risks catastrophic forgetting for previous stage. Furthermore, independent optimization ignores the dependency between the two tasks, leading to error propagation. Therefore, we propose a joint learning framework inspired by the Information Bottleneck (IB) principle. We treat the extracted reasoning logic graph $\mathcal{G}$ not merely as an intermediate output, but as a compressed semantic bottleneck that distills the essential information (Tishby & Zaslavsky, 2015; Tishby et al., 2000) from the paper body $\mathcal{B} = (\mathcal{M}, \mathcal{R}, \mathcal{A}, \mathcal{C})$ required to reconstruct the introduction $\mathcal{I}$, where $\mathcal{M}, \mathcal{R}, \mathcal{A}, \mathcal{C}$ are the methodology, result, analysis and citation section of the paper, respectively.

Instead of being supervised by static labels, we cast the entire trajectory $\mathcal{B} \rightarrow \mathcal{G} \rightarrow \mathcal{I}$ as a single unified episode within an RL paradigm. The model is trained to maximize a set of **carefully designed fine-grained rewards**. These rewards evaluate the quality of the final generated Introduction, providing feedback that propagates back to optimize the generation and extraction policies simultaneously.

### 3.3.1. SIMPLIFIED REINFORCEMENT LEARNING

Drawing inspiration from the efficiency of Group Relative Policy Optimization (GRPO) (Shao et al., 2024), we propose a *Simplified PPO* architecture tailored for our joint extraction-generation task. To reduce memory overhead and training cost, we streamline the system to encompass only two active components: a **Policy Model** (Actor, $\pi_\theta$) and a **Value Model** (Critic, $V_\phi$).

Specifically, we discard the learned Reward Model, as the quality of logic extraction and text generation in our domain can be evaluated through deterministic rules rather than black-box predictions. Consequently, the training objective is driven by a *verifiable reward* function. Let a trajectory be $\tau = (\mathcal{B}, \mathcal{G}, \mathcal{I})$. The optimization objective is defined as:

$$\mathcal{L}^{CLIP}(\theta) = \mathbb{E}_{\tau \sim \pi_\theta} \left[ \min \left( \rho_t(\theta)\hat{A}_t, \text{clip}(\rho_t(\theta), 1 - \epsilon, 1 + \epsilon)\hat{A}_t \right) \right],$$
(4)

where $\rho_t(\theta) = \frac{\pi_\theta(a_t|s_t)}{\pi_{\theta_{old}}(a_t|s_t)}$ is the probability ratio, and $\hat{A}_t$ is the advantage estimated by the Critic $V_\phi$. Crucially, the advantage calculation relies on our verifiable reward $R(\tau)$, which is formulated as a weighted sum of feedback signals:

$$R(\tau) = R_{\text{graph}}(\mathcal{G}) + R_{\text{faith}}(\mathcal{G}, \mathcal{I}) + R_{\text{consis}}(\mathcal{I}) + R_{\text{qual}}(\mathcal{I}) + R_{\text{ref}}(\mathcal{I}),$$
(5)

where $\mathcal{G}$ and $\mathcal{I}$ denote the extracted reasoning graph and the generated introduction within the trajectory $\tau$, respectively. Specifically, $R_{\text{graph}}$ acts as a structural regularizer for the intermediate representation, $R_{\text{faith}}$ penalizes hallucinations to ensure the text strictly follows the graph, $R_{\text{ref}}$ provides supervised guidance from the ground truth, and $R_{\text{qual}}$ enforces high-level academic writing standards. The detailed design of the rewards is as follows.

### 3.3.2. VERIFIABLE REWARD MODELING

To steer the model toward generating logically sound and rigorous introductions, we design a composite reward function $R(\tau)$ aggregated from five distinct dimensions: graph validity, generation faithfulness, paper consistency, academic quality, and reference alignment.

**Graph Validity** ($R_{\text{graph}}$) ensures the intermediate reasoning graph $\mathcal{G}$ is topologically valid and informative. We employ *Reasoning Edge Accuracy*, where an LLM verifier checks if the premise node logically supports the conclusion node for each edge, defining the reward as the ratio of validated edges. Additionally, we compute *Entity Coverage* by measuring the overlap between entities in $\mathcal{G}$ and key concepts extracted from the ground-truth introduction $\mathcal{I}^*$, encouraging the graph to capture essential research concepts.

**Faithfulness Rewards** ($R_{\text{faith}}$). Since $\mathcal{I}$ is generated solely from $\mathcal{G}$, strict adherence to the graph's semantics is critical to prevent hallucination. We enforce this via *Bidirectional Coverage*, which penalizes ungrounded content by calculating the semantic overlap of key phrases between $\mathcal{G}$ and $\mathcal{I}$. Furthermore, we assess *Entailment Faithfulness* using the `SummaC` model to compute NLI scores (treating the linearized graph as the premise), and measure *Contextual Relevance* via the cosine similarity between the embeddings of the graph and the generated text.

**Paper Consistency** ($R_{\text{consis}}$). Using the original introduction $\mathcal{I}^*$ as a proximal reference, we guide the optimization using supervised signals. We calculate *Lexical and Semantic Similarity* via BLEU scores and dense vector embeddings from an embedding LLM (e.g., Qwen3-Embedding). To ensure the recovery of core arguments, we also evaluate *Key Point Consistency*, which measures the recall of key phrases and logical entailment against the ground truth $\mathcal{I}^*$.

**Academic Quality** ($R_{\text{qual}}$). To capture high-level nuances of scientific writing, we deploy an LLM-as-a-Judge evaluator. This module scores the generation on normalized scales regarding *Coherence*, *Completeness*, and *Academic Tone*, where an LLM is prompted to generate these scores. Finally, we include an *Entirety Preference* signal, a binary reward indicating if the generated introduction is qualitatively comparable to or strictly better than $\mathcal{I}^*$.

**Reference Alignment** ($R_{\text{ref}}$). Moreover, we evaluate *Citation Integrity* by checking the recall and contextual usage correctness of references.

## 4. Experiments

### 4.1. Experimental Setup

**Datasets.** We construct a large-scale scientific paper dataset from *Nature Communications*, motivated by the observa-

*Table 2.* Main experimental results comparing our method against strong proprietary LLMs and baselines. **GQ**: Graph Quality, **GW**: Graph-Writing Alignment, **PC**: Paper Consistency, **WQ**: Writing Quality, **CQ**: Citation Quality, **OP**: Overall Performance. The *One-Step-Baseline* lacks intermediate graph generation, hence GQ and GW are not applicable.

| Model | GQ↑ | GW↑ | PC↑ | WQ↑ | CQ↑ | OP↑ |
|---|---|---|---|---|---|---|
| *Proprietary SOTA Models* | | | | | | |
| GLM-4.7 (GLM-4.7Team, 2025) | 0.123 | 0.088 | 0.428 | 0.734 | **0.739** | 0.466 |
| Gemini-2.5pro (Comanici et al., 2025) | 0.357 | 0.262 | 0.458 | 0.814 | 0.710 | 0.566 |
| Grok4 (Grok4Team, 2025) | 0.651 | 0.601 | 0.464 | 0.691 | 0.721 | 0.599 |
| Claude-haiku-4.5 (Anthropic, 2025) | 0.707 | **0.727** | 0.470 | 0.699 | 0.529 | 0.612 |
| GPT-o3 (OpenAI, 2025) | 0.690 | 0.448 | 0.416 | **0.882** | 0.691 | 0.656 |
| *Our Framework (Backbone: Qwen3-4B)* | | | | | | |
| Qwen3-4B-Instruct-2507 (Base) | 0.478 | 0.682 | 0.453 | 0.546 | 0.444 | 0.510 |
| One-Step-Baseline | – | – | 0.476 | 0.829 | 0.477 | – |
| **LECTOR (Our Method)** | **0.745** | 0.623 | **0.486** | 0.834 | 0.530 | **0.665** |

tion that high-quality scientific articles exhibit rich and explicit research reasoning structures. Specifically, we collect 10,200 peer-reviewed papers spanning multiple subfields, including *Astronomy and Planetary Science*, *Energy Science and Technology*, *Materials Science*, *Nanoscience and Technology*, *Physics*, *Chemistry*, *Engineering*, *Mathematics and Computing*, and *Optics and Photonics*. The publication dates of the collected papers range from April 2010 to March 2025, covering more than a decade of scientific developments. For each paper, we use MinerU (Wang et al., 2024a) to automatically parse the PDF files and extract structured sections. Following our task definition, the Introduction section is excluded from the input and reserved as the target output for evaluation, while the remaining main body sections are used for reasoning logic graph extraction. The dataset is randomly split into training, validation, and test sets containing 10,000, 100, and 100 papers, respectively.

**Evaluation Metrics.** We evaluate model performance from both reasoning and writing perspectives using five complementary metrics: Graph Quality (GQ, correctness and completeness of extracted reasoning graphs), Graph-Write Alignment (GW, alignment between graphs and generated Introductions), Paper Consistency (PC, factual and semantic consistency with source papers), Writing Quality (WQ, fluency, coherence, and academic writing quality), and Citation Quality (CQ, correctness and completeness of citations). We further report an Overall Performance (OP) score by aggregating all metrics. All scores are normalized to $[0, 1]$. Detailed definitions are provided in Appendix A.

**Implementation Details.** We utilize Qwen3-4B-Instruct-2507 (Yang et al., 2025) as the backbone model, with the reinforcement learning pipeline built upon the `verl-agent` framework. For reward modeling and *LLM-as-a-judge* assessments, we leverage the capabilities of the larger Qwen3-235B model. Auxiliary metrics rely on specialized encoders: `Qwen3-Embedding-0.6B` and `all-MiniLM-L6-v2` are employed for semantic vector representations, while logical entailment is scored using `mnli-base` (via the SummaC protocol). Key phrases are extracted using the

YAKE algorithm. Optimization proceeds for a single epoch with a global batch size of 64 and a learning rate of $1 \times 10^{-6}$.

## 4.2. Main Results

Table 2 compares our proposed framework against both strong proprietary LLMs and baseline approaches. Initially, the base model (*Qwen3-4B-Instruct-2507*) exhibits a noticeable performance gap compared to commercial giants like GPT-o3 (OpenAI, 2025) and Claude-haiku-4.5 (Anthropic, 2025), highlighting the inherent challenge of logic-aware scientific writing under limited parameter capacity.

Upon applying our joint RL training, the model achieves substantial improvements across nearly all dimensions. Most notably, **Graph Quality (GQ)** surges by $+0.267$ (from 0.478 to 0.745) and **Writing Quality (WQ)** improves by $+0.288$ (from 0.546 to 0.834). These gains confirm that our dual-objective optimization effectively creates a positive feedback loop: better logic extraction facilitates clearer writing, which in turn reinforces the extraction of salient reasoning paths. Remarkably, despite utilizing a significantly smaller backbone (4B parameters), our method outperforms larger proprietary models like Grok4 (Grok4Team, 2025) and Gemini-2.5pro (Comanici et al., 2025) in **Overall Performance (OP)**, and even achieves parity with GPT-o3 (OpenAI, 2025) (0.665 vs. 0.656). This demonstrates the high parameter efficiency of our specialized RL training.

We observe a slight decline in Graph-Write Alignment (GW), dropping from 0.682 to 0.623. Rather than a failure, we conjecture this as a shift in the generation strategy: while the base model tends to perform rigid node-to-text translation, the RL-optimized model prioritizes *discourse fluency* and *semantic integration*. By relaxing strict lexical alignment, the model learns to synthesize the graph's logical skeleton into more natural, coherent prose, as evidenced by the dramatic rise in Writing Quality.

Comparing LECTOR to the *One-Step-Baseline* (direct generation without intermediate graphs) reveals the critical role

*Table 3.* Ablation study on the effects of reasoning logic graph on Introduction writing. † denotes a same LECTOR model for graph generation and introduction writing (i.e., the proposed framework).

| Graph Gen. Model | Writing Model | GQ↑ | GW↑ | PC↑ | WQ↑ | CQ↑ | OP↑ |
|---|---|---|---|---|---|---|---|
| Qwen3 | Qwen3 | 0.478 | 0.682 | 0.453 | 0.546 | 0.444 | 0.510 |
| **LECTOR** | Qwen3 | **0.745** | **0.637** | 0.473 | 0.640 | 0.480 | 0.568 |
| Qwen3 | **LECTOR** | 0.478 | 0.601 | 0.476 | 0.808 | 0.515 | 0.615 |
| **LECTOR†** | **LECTOR†** | **0.745** | 0.623 | **0.486** | **0.834** | **0.530** | **0.665** |

*Table 4.* Ablation study on joint training. We compare LECTOR with two separated stages: **Separated-Step1**† is trained solely on graph extraction, and **Separated-Step2**‡ is trained on Introduction writing using graphs generated by Separated-Step1.

| Model | GQ↑ | GW↑ | PC↑ | WQ↑ | CQ↑ | OP↑ |
|---|---|---|---|---|---|---|
| Qwen3 | 0.478 | 0.682 | 0.453 | 0.546 | 0.444 | 0.510 |
| LECTOR | 0.745 | 0.623 | 0.486 | 0.834 | 0.530 | 0.665 |
| Separated-Step1† | 0.943 | 0.679 | 0.463 | 0.662 | 0.500 | 0.603 |
| Separated-Step2‡ | 0.495 | 0.638 | 0.458 | 0.814 | 0.432 | 0.618 |

of the information bottleneck. Our method surpasses the baseline in **Paper Consistency** (0.486 vs. 0.476) and **Citation Quality** (0.530 vs. 0.477). This suggests that explicitly modeling the reasoning logic graph serves as an effective cognitive scaffold, enabling the model to better organize complex scientific arguments and ground citations than a black-box end-to-end approach.

## 4.3. Ablation Study

**Impact of Reasoning Logic Graph Quality.** To isolate the contribution of the intermediate logic graph, we decouple the extraction and generation stages, evaluating four cross-combinations of the baseline and our RL-trained model (*LECTOR*). The results are presented in Table 3.

First, fixing the writer to the base model (Rows 1 vs. 2) reveals that upgrading the graph extractor alone yields substantial gains across all metrics. Specifically, replacing the base graph with the *LECTOR*-extracted graph boosts Graph Quality (GQ) by +0.267, which cascades into improvements in Paper Consistency (+0.020) and Writing Quality (+0.094). This confirms that *a superior intermediate representation is a prerequisite for high-fidelity generation*.

Second, when the writer is also upgraded to *LECTOR* (Rows 3 vs. 4), the benefit of a high-quality graph becomes even more pronounced. The transition from a base graph to a *LECTOR* graph in this setting further lifts Writing Quality (+0.026) and Citation Quality (+0.015). Comparing the full pipeline (Row 4) against the partially optimized settings demonstrates a clear synergistic effect: *the joint optimization ensures that the graph extractor learns to capture information specifically tailored to the writer's needs*, maximizing the overall performance (OP) to 0.665.

**Effects of Joint Optimization.** To evaluate the efficacy of joint optimization, we compare our unified LECTOR model

*Table 5.* Ablation study on individual reward components. We compare variants of our model by removing GQ, GW, PC, WQ, and CQ rewards individually against the full LECTOR framework.

| Remove Reward | GQ↑ | GW↑ | PC↑ | WQ↑ | CQ↑ | OP↑ |
|---|---|---|---|---|---|---|
| LECTOR | 0.745 | 0.623 | 0.486 | 0.834 | 0.530 | 0.665 |
| - GQ | 0.408 | 0.624 | 0.466 | 0.815 | 0.513 | 0.618 |
| - GW | 0.719 | 0.615 | 0.464 | 0.823 | 0.455 | 0.642 |
| - PC | 0.775 | 0.635 | 0.460 | 0.806 | 0.617 | 0.653 |
| - WQ | 0.890 | 0.931 | 0.453 | 0.258 | 0.588 | 0.500 |
| - CQ | 0.755 | 0.616 | 0.464 | 0.824 | 0.311 | 0.634 |

*Table 6.* Human evaluation results. 8 domain experts scored each system on four dimensions (1–5 scale) and provided overall rankings.

| Dimension | Original | Base | GPT-o3 | LECTOR |
|---|---|---|---|---|
| Logical Coherence | 3.74 | 2.51 | 4.00 | **4.05** |
| Writing Quality | 3.54 | 2.51 | **4.12** | 3.91 |
| Citation Integration | 3.21 | 2.06 | **3.51** | 2.99 |
| Completeness | 3.59 | 2.46 | 3.21 | **3.99** |
| Overall | 3.52 | 2.38 | 3.71 | **3.73** |
| Ranked 1st | 10.6% | 0% | **48.8%** | 40.6% |

against independent training stages, *i.e.*, *Separated-Step1* trained on graph extraction, *Separated-Step2* is trained on Introduction writing using graphs generated by Separated-Step1. As shown in Table 4, training the graph extractor alone (*Separated-Step1*) still improves writing metrics over the baseline, increasing Paper Consistency by +0.010, Writing Quality by +0.116, and Citation Quality by +0.056, confirming that enhanced reasoning structures benefit generation even without writing-specific tuning. However, isolated optimization of the writing module (*Separated-Step2*) fails to compensate for degraded reasoning inputs, leading to lower Writing Quality (-0.020) and Citation Quality (-0.098) relative to LECTOR. These results demonstrate that joint training is critical for LECTOR, as it preserves the quality of reasoning representations while optimizing for coherent, evidence-grounded Introduction generation.

**Effects of Reward Modules.** To analyze the contribution of individual reward components, we remove one reward at a time while keeping other settings constant. As shown in Table 5, omitting any reward component leads to a clear degradation in its Overall Performance and corresponding target metric, most notably for Writing Quality (0.834 → 0.258) and Citation Quality (0.530 → 0.311). This confirms that each reward uniquely guides the model toward specific optimization goals. Second, removing certain rewards also leads to degradation in other metrics, reflecting the synergistic nature of our framework. Specifically, removing the Graph-Writing alignment (GW) reward weakens the grounding of text, reducing Citation Quality (CQ) from 0.530 to 0.455, while omitting the Graph Quality (GQ) reward adversely impacts downstream Writing Quality (WQ) from 0.834 to 0.815. These results demonstrate that the full suite

| Original | GPT-o3 | LECTOR (Ours) |
|---|---|---|
| *The Kagome lattice of chiral AFM Mn₃X also displays a large AHE that arises from the non-zero Berry phase...* 
 ... 
 *...Moreover, Mn₃Sn and MnGe exhibit different scaling relationships...* 
 *...The two-spin correlation is a good method to explain the large intrinsic AHE in Mn₃Sn, by the vector spin chirality...* 
 ... 
 *...Besides intrinsic AHE for chiral AFM, skew scattering topological AHE can be induced by the three-spin correlated scalar spin chirality...* 
 ... 
 *Here, we systematically study the AHE of the chiral non-collinear AFM Mn₃Pt and find a universal AHE scaling law.* | *The AHE has emerged as a key probe in modern spin-orbitronics... Two microscopic mechanisms are known to contribute...* 
 *...First, scalar spin chirality introduces a skew-scattering channel...* 
 *...Second, the vector spin chirality gives rise to an intrinsic Berry-curvature term...* 
 ... 
 *...a quantitative framework capable of unifying the extrinsic and intrinsic contributions has remained elusive.* 
 ... 
 *no universal law has yet been established that simultaneously captures (i)... and (ii)...* 
 ... 
 *Here we demonstrate that the requisite unification emerges naturally...* | *Previous studies have established that both VSC and SSC contribute to the AHE, with the intrinsic component originating from Berry curvature...* 
 *...Experimental observations in Mn₃Ge, Mn₃Sn, and Mn₃Ir have confirmed the presence of large AHEs...* 
 ... 
 *A critical gap in the current understanding lies in the lack of a universal scaling law that can simultaneously account for...* 
 ... 
 *the physical origin of this consistency has not been established.* 
 ... 
 *To address this gap, we propose a unified framework...* 
 $\rho_{AH} = a_{sk}\rho_{xx} + b_{in}\rho_{xx}^2$...*Hall conductance from 56.73 to 73.90 $\Omega^{-1}$cm$^{-1}$...* |

*Figure 3.* Qualitative comparison of original, GPT-o3-generated, and LECTOR-generated Introductions. Colors denote Swales' CARS rhetorical moves: Territory , Niche , and Contribution . Complete case studies are provided in Appendix E.

of rewards is essential for generating high-quality reasoning representations and logically faithful introduction writing.

### 4.4. Human Evaluation

To validate the reliability of our automatic evaluation framework, we conduct a comprehensive human evaluation with 8 domain experts on 20 randomly sampled test papers. Each expert independently scored LECTOR, the Base model (Qwen3-4B zero-shot), GPT-o3, and the Original (ground-truth) Introduction across four dimensions—Logical Coherence, Writing Quality, Citation Integration, and Completeness—on a 1–5 scale, and also provided an overall ranking. Results are shown in Table 6.

LECTOR achieves the highest overall score (3.73) and excels in Logical Coherence (4.05) and Completeness (3.99), reflecting the benefit of explicit reasoning graph structure. Notably, LECTOR and GPT-o3 achieve comparable overall scores (3.73 vs. 3.71), with both systems receiving over 40% of first-place rankings (40.6% vs. 48.8%), yet with complementary strengths: GPT-o3 leads on Writing Quality and Citation Integration, while LECTOR leads on Logical Coherence and Completeness. This is consistent with the automatic evaluation results in Table 2.

Moreover, the Spearman correlation between human and LLM judgments is $\rho = 0.815$ ($p < 0.001$), with Krippendorff's $\alpha = 0.758$, confirming substantial agreement. These results validate the reliability of our LLM-as-Judge evaluation framework and confirm that LECTOR's improvements reflect genuine quality gains.

### 4.5. Qualitative Case Study

To qualitatively assess the structural quality of generated Introductions, we compare outputs from the Original, GPT-o3, and LECTOR on representative test papers. Figure 3 shows annotated excerpts from one case, with colors denoting Swales' CARS rhetorical moves. Three complete case studies covering different physics subdomains are provided in Appendix E (Figures 6–8).

Across all three case studies, we observe consistent patterns. First, LECTOR produces a clear Swales' CARS structure (Territory→Niche→Contribution), while the Originals mix background and contributions in single dense paragraphs without explicit rhetorical transitions. Second, LECTOR explicitly states research gaps (e.g., "A critical gap in the current understanding lies in..."), whereas the Originals pose only implicit questions or abruptly introduce contributions. Third, compared to GPT-o3 which generates polished but high-level prose, LECTOR includes more technical depth—for instance, the explicit AHE scaling law $\rho_{AH} = a_{sk}\rho_{xx} + b_{in}\rho_{xx}^2$ and first-principles Hall conductance calculations in Case Study 1 (Figure 6), explicit Bloch Hamiltonian and second Chern number formulas ($C_2 = 3$) in Case Study 2 (Figure 7), and quantitative XAFS fitting results (Se–Nb bond elongation of 0.02 Å) in Case Study 3 (Figure 8).

## 5. Conclusion

In this paper, we investigate the challenge of scientific introduction writing, a pivotal frontier in the development of AI Scientist systems. We suggest that this task is fundamentally a task of structured reasoning rather than simple text generation. To address this problem, we introduce the **Content-Conditioned Introduction Generation (CCIG)** task and the **LECTOR** framework, which anchors narratives in technical substance via a **Logic-Reasoning Graph**. Through extensive experiments and evaluations, we demonstrate that LECTOR is highly effective, enabling a lightweight 4B-parameter model to match or surpass the logical fidelity of state-of-the-art commercial models. By synchronizing logical fidelity with narrative expression, this work advances the frontiers of trustworthy, verifiable AI research assistants.

## Acknowledgements

This work was supported by the JC STEM Lab of AI for Science and Engineering, funded by The Hong Kong Jockey Club Charities Trust, the MTR Research Funding (MRF) Scheme (CHU-24003), and the Research Grants Council of Hong Kong (Project No. CUHK14213224).

## Impact Statement

This paper presents work aimed at making scientific AI systems more **interpretable, verifiable, and logically grounded**, which, in our view, yields a significant positive societal impact. Most current research on AI-assisted scientific writing treats the generation of complex research narratives as a standard surface-level text-completion task. This has serious societal and academic drawbacks, as "black-box" models often prioritize linguistic fluency over factual accuracy, leading to **AI hallucinations**—such as the fabrication of citations and the creation of logically disconnected arguments. By failing to model the underlying logic of a discovery, current LLMs risk polluting the scientific record with plausible-sounding but groundless content, a trend that undermines the foundation of empirical research.

Our work addresses this by introducing the **Content-Conditioned Introduction Generation (CCIG)** task and the **LECTOR** framework. By forcing the model to first construct an explicit **Logic-Reasoning Graph**, we shift the paradigm from mere text mimicry to **structured deduction**. This ensures that the generated narrative is a faithful reflection of the actual methodology and results, thereby increasing the transparency and reliability of AI-assisted scientific communication.

Since LECTOR significantly improves the quality of scientific writing, concerns may arise regarding its potential misuse by "paper mills" for large-scale fraudulent manuscript generation. However, we argue that our framework actually **increases the barrier to deceptive automation** in two critical ways:

- **High-Fidelity Constraints:** Unlike general-purpose generators, LECTOR requires a high-quality, rigorous set of research materials (Methodology, Results, and Analyses) to construct a valid Logic-Reasoning Graph. This dependency ensures that the system cannot easily produce high-quality narratives out of "thin air" without substantive underlying research.

- **Structural Verifiability:** Because the generation is conditioned on an explicit logical blueprint, any attempt to fabricate a paper requires the fabrication of an entire coherent logical structure. Such forged structures are inherently more fragile and easier for human experts or automated auditing tools to detect compared

to the subtle, fluid hallucinations of black-box models.

By prioritizing **logic fidelity** over mere topical fluency, LECTOR provides the scientific community with a more principled and accountable pathway for integrating generative AI into the research workflow, ensuring that technology serves to uphold, rather than compromise, intellectual rigor.

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

# A. Evaluation Metrics

We design a comprehensive evaluation protocol to assess both reasoning logic quality and Introduction writing quality. All metrics are normalized to the range $[0, 1]$ for consistency. Metrics are organized into five primary groups: Graph Quality (GQ), Graph-Write Alignment (GW), Paper Consistency (PC), Writing Quality (WQ), and Citation Quality (CQ).

**Graph Quality (GQ).**   GQ evaluates the correctness and completeness of the extracted reasoning logic graph, computed as the average of:

- **Reasoning Edge Accuracy (REA):** The proportion of reasoning edges validated as logically sound by an LLM-based verifier.

- **Entity Coverage (EC):** The proportion of core scientific entities from the reference captured by the graph.

**Graph-Write Alignment (GW).**   GW measures the grounding of the generated Introduction in the reasoning graph, computed as the average of:

- **Contextual Relevance:** Semantic similarity between graph node text and the Introduction.

- **Graph Coverage:** The proportion of graph-derived key phrases covered by the Introduction.

- **Key Phrase Faithfulness:** The proportion of Introduction key phrases grounded in the graph.

- **Entailment Faithfulness:** NLI-based entailment score between the graph and the Introduction.

**Paper Consistency (PC).**   PC evaluates the factual alignment with the original paper, computed as the average of:

- **Lexical Similarity:** BLEU score relative to the reference Introduction.

- **Semantic Similarity:** Embedding-based similarity using Qwen3-Embedding.

- **Paper Coverage:** Recall of reference key phrases in the generated text.

- **Key Phrase Consistency:** Precision of generated key phrases against reference phrases.

- **Entailment Consistency:** NLI-based consistency between reference and generated Introductions.

**Writing Quality (WQ).**   WQ evaluates high-level academic writing quality using LLM-based judges. This group consists of 11 distinct dimensions. Ten dimensions are scored on a Likert scale of 1–5, while **Preference** is a binary metric. For consistency, all scores are normalized to the range $[0, 1]$. The WQ score is the arithmetic mean of the following aspects:

- **Consistency with Original Introduction:** Whether the generated content is logically consistent with the source without contradictions.

- **Coverage of Key Points:** Whether core ideas, arguments, and contributions are sufficiently captured.

- **Background and Context Quality:** The adequacy and appropriateness of the provided background information.

- **Problem Clarity:** The precision and clarity of the research problem definition.

- **Motivation and Significance:** How convincingly the importance and necessity of the research are explained.

- **Related Work Positioning:** Whether the work is properly situated within the existing scientific literature.

- **Contribution Clarity:** Whether the main contributions are explicitly and clearly stated.

- **Logical Structure:** Adherence to standard academic organizational structures.

- **Coherence and Flow:** The logical connectivity and smoothness of the discourse.

- **Academic Writing Quality:** General adherence to professional academic writing standards.

- **Preference (Binary):** A binary judgment indicating whether the generated Introduction exhibits superior or equal overall quality compared to the reference.

**Citation Quality (CQ).** CQ evaluates citation usage, computed as the average of:

- **Reference Recall:** Recall of cited sources against the reference Introduction.

- **Reference Usage Correctness:** LLM-evaluated appropriateness of citation contexts.

**Overall Performance (OP).** To provide a unified evaluation, we compute the **Overall Performance (OP)** as the arithmetic mean of all $N$ individual sub-metrics across the five groups ($N = 24$, comprising 2 from GQ, 4 from GW, 5 from PC, 11 from WQ, and 2 from CQ):

$$OP = \frac{1}{N} \sum_{m \in \mathcal{M}} m$$

This holistic metric ensures that the joint optimization of reasoning logic and generative quality is captured with fine-grained sensitivity.

## B. Detailed Prompt Specifications

To implement our **LECTOR** framework, we carefully designed two specialized prompts: the **Reasoning Logic Graph Extraction** prompt and the **Logic-Aware Introduction Writing** prompt. These are presented in Figure 4 and Figure 5, respectively.

- The **Reasoning Logic Graph Extraction** prompt is designed to operationalize the logic abstraction stage in LECTOR by converting unstructured scientific text into a verifiable symbolic representation. Specifically, the prompt formalizes Peirce's three reasoning paradigms (deduction, abduction, and induction) and enforces a strict *edge pairing constraint*, which requires every inferred conclusion to be supported by exactly two complementary premises corresponding to its reasoning type. This constraint prevents under-specified or heuristic inferences and forces the model to explicitly instantiate the logical justification behind each claim. In addition, the prompt requires all nodes to be expressed as atomic, self-contained declarative sentences and constrains the output to a single-rooted Graphviz DOT reasoning tree. These design choices ensure that the extracted graph serves as a faithful logical blueprint of the paper, explicitly exposing intermediate reasoning steps, preserving multi-hop dependency structures, and enabling downstream modules to reason over a structured, interpretable representation rather than raw text.

- The **Logic-Aware Introduction Writing** prompt reformulates Introduction generation as a structure-guided realization problem rather than a free-form text generation task. It explicitly binds the generated content to the extracted reasoning graph by requiring the model to preserve all nodes and reasoning relations when producing the Introduction. At the discourse level, the prompt enforces Swales' CARS rhetorical framework, constraining the narrative flow to follow the canonical sequence of territory establishment, niche identification, and contribution presentation. At the grounding level, it strictly restricts citations to the provided reference list using fixed index-based markers, eliminating unsupported references and citation hallucinations. By jointly constraining logical content, rhetorical structure, and citation grounding, this prompt ensures that the generated Introduction remains logically consistent with the underlying paper structure while satisfying academic writing conventions.

---

**Prompt for Reasoning Logic Graph Extraction**

Charles S. Peirce, a member of the National Academy of Sciences of the United States, pointed out that all valid reasoning is either deductive, inductive, or hypothetic; or else it combines two or more of these characters.

Now, I provide the main body of a scientific article (excluding the Introduction section). Please extract its core scientific research proposal or idea, and use the above three types of reasoning to explicitly reconstruct the reasoning process leading to that idea. Please output the complete logical reasoning chain in Graphviz DOT syntax as a single graph. Your output must consist of the complete Graphviz DOT graph only. Do NOT include any explanations, comments, natural language descriptions, or any text outside the DOT code block.

Please strictly abide by the following requirements: Overall goal: Extract the logical structure behind the scientific research from the original text. Its structure should reflect the end-to-end logical organization of the entire paper, so that the resulting reasoning graph can serve as a structural basis for writing the Introduction section. The graph describes how the paper's core research idea is motivated, justified, and formed from the main body content. Specifically, you will build a single-rooted reasoning tree, and the following is the specific definition.

1. Each node should be written as a complete sentence (Transcription) that explicitly reflects a step in the reasoning process.
The information expressed in the Transcription may originate from one of the following situations: - An original sentence from the paper - An original viewpoint or claim inferred from the paper - An opinion or statement derived from referenced work - Implicit information or background knowledge used for reasoning
You do NOT need to explicitly label which situation a node belongs to. However, each node must correspond to exactly one reasoning unit or atomic piece of information. Do not combine multiple independent reasoning units into a single node. If multiple pieces of information are needed, create separate nodes and connect them through reasoning edges.
2. The edge type can only be one of the following 6 types: deduction-rule, deduction-case, abduction-phenomenon, abduction-knowledge, induction-case, induction-common
3. CRITICAL CONSTRAINT - Edge Pairing Requirements: Every reasoning conclusion must be reached by exactly two edges of specific paired types pointing to the same target node. The valid pairs are: - For deductive reasoning: One "deduction-rule" edge and one "deduction-case" edge must both point to the same target node - For abductive reasoning: One "abduction-phenomenon" edge and one "abduction-knowledge" edge must both point to the same target node - For inductive reasoning: One "induction-case" edge and one "induction-common" edge must both point to the same target node
This means if you have a conclusion node reached by deductive reasoning, there must be exactly one incoming "deduction-rule" edge from one source node and exactly one incoming "deduction-case" edge from another source node, both targeting the same conclusion node.
4. Other constraints: a. If there is multi-hop reasoning in the reasoning chain (or a single reasoning/induction/deduction is not enough to explain clearly, such compound reasoning is common in scientific literature), please introduce intermediate nodes (as implicit information) to break down the logical path into multiple clear reasoning steps. The intermediate nodes must also be written as complete sentences (Transcription). The more detailed the reasoning and the more nodes there are, the higher the score.
b. Domain consensus can be used as implicit information, but it cannot be written directly in the form of a conclusion, and must be written as callable background knowledge.
c. A single node may serve simultaneously as the conclusion of one reasoning step and as the argument (premise) of a subsequent reasoning step. In such cases, use a single Transcription for the node and connect it to different reasoning edges according to its roles.
d. The reasoning tree must have exactly one root node with input edges only. This root node represents the determination of the core scientific research idea or method, summarizing the overall research logic of the entire paper in a form that can be naturally articulated and motivated in the Introduction section.
e. You should aim to construct a connected, multi-step Reasoning Tree that gradually leads to this root node. Avoid linking most nodes directly to the root node; instead, intermediate conclusions should be reused as premises for subsequent reasoning steps, forming deeper and more coherent reasoning paths. All nodes must be part of this logical backbone, with no abandoned or isolated nodes.
f. Graph size constraint: The total number of nodes in the graph must NOT exceed 50. If the reasoning process would naturally exceed this limit, you must prioritize the most essential reasoning steps, merge highly similar or redundant intermediate nodes when logically possible, and preserve the main multi-hop reasoning backbone that leads to the root node.
PAPER CONTENT: {Paper Content}

*Figure 4.* Prompt for Reasoning Logic Graph Extraction.

## Prompt for Logic-Aware Introduction Writing

You are given a Graphviz DOT file that encodes the complete logical reasoning structure behind a scientific research idea. This DOT graph represents a single-rooted reasoning tree derived from the main body of a scientific article, capturing the end-to-end logical structure of the entire paper. The graph is intended to serve as the structural and logical basis for writing the Introduction section. Each node corresponds to a specific piece of information (original sentence, referenced opinion, or implicit knowledge), and each pair of edges represents a reasoning step following Peirce's three modes of reasoning: deduction, abduction, and induction.
In addition to the DOT graph, you are also given a references list. The references list contains bibliographic entries that may be cited in the Introduction, each associated with a unique index.
Your task is to use the reasoning graph as the sole source of scientific content and the provided references list as the sole source of citations, and write a complete Introduction section for a scientific paper.
You must fully preserve, use, and cover all concepts, relations, and reasoning steps contained in the DOT graph, translating them into coherent academic prose, and insert citations at appropriate positions where prior work, background knowledge, or related studies are involved.
All citations MUST strictly follow this format: [idx], where idx is the index of the corresponding reference in the given references list. Do not invent, modify, or omit reference indices.
—
Swales' CARS Model Requirement:
The Introduction must explicitly follow Swales' CARS (Create A Research Space) model, which consists of three rhetorical moves:
1. Move 1 — Establishing the territory: Describe the broad research area and its importance. Summarize the established background knowledge relevant to the DOT graph.
2. Move 2 — Establishing the niche: Identify gaps, unresolved issues, limitations, or unanswered questions implied by the reasoning

graph.
3. Move 3 — Occupying the niche: Present the central research idea, method, or proposal (corresponding to the root node of the graph), showing how it logically follows from the reasoning chain.
Your Introduction must integrate all nodes, reasoning relations, and required citations into a clear, natural, and academically polished narrative.

—

Writing Requirements:
- Use clear, natural, academic English appropriate for a top-tier scientific journal. - Ensure the Introduction fully reflects the logical structure encoded in the Graphviz DOT file. - Ensure all citations come exclusively from the provided references list and use the required [idx] format. - Use Markdown code formatting for all math symbols and equations. - Do not omit any important information represented anywhere in the graph. - DO NOT include commentary, explanations of the DOT file, or any text outside the required structure. - DO NOT add references that are not explicitly provided.

—

Output Format (MANDATORY):
Your output must include only the Introduction content. Do not include any titles, headings, sections, notes, or explanations—just the text of the Introduction itself.

—

Final Instruction:
Now, given the following Graphviz DOT code and the corresponding references list, write the required Introduction.
GRAPHVIZ DOT: {Reasoning Logic Graph}
REFERENCES: {Citation List}

*Figure 5.* Prompt for Logic-Aware Introduction Writing.

## C. Statistical Significance Analysis

To ensure the robustness of our experimental results, we conduct comprehensive statistical analyses on all 100 test papers using paired per-paper comparisons (LECTOR vs. Base). Table 7 reports 95% bootstrap confidence intervals (10,000 resamples), Holm–Bonferroni corrected $p$-values from paired $t$-tests, and Cohen's $d_z$ effect sizes.

*Table 7.* Statistical significance of LECTOR vs. Base across all metrics. All improvements are significant after Holm–Bonferroni correction ($p < 3 \times 10^{-6}$) with large effect sizes.

| Metric | LECTOR [95% CI] | Base [95% CI] | Δ | Holm $p$ | Cohen's $d_z$ |
|---|---|---|---|---|---|
| GQ | 0.745 [0.701, 0.784] | 0.478 [0.427, 0.526] | +0.267 | 1.45e-11 | 0.866 |
| PC | 0.486 [0.474, 0.498] | 0.453 [0.440, 0.467] | +0.032 | 2.07e-08 | 0.675 |
| WQ | 0.834 [0.814, 0.854] | 0.546 [0.518, 0.573] | +0.289 | 5.89e-16 | 1.592 |
| CQ | 0.530 [0.500, 0.560] | 0.444 [0.419, 0.468] | +0.086 | 2.91e-06 | 0.517 |
| OP | 0.665 [0.651, 0.679] | 0.510 [0.498, 0.522] | +0.155 | 5.89e-16 | 1.617 |

All improvements are statistically significant after Holm–Bonferroni correction (all $p < 3 \times 10^{-6}$), with large effect sizes for Writing Quality ($d_z = 1.592$) and Overall Performance ($d_z = 1.617$), and non-overlapping 95% bootstrap confidence intervals across all metrics. Moreover, the statistical power is sufficient with all $p < 10^{-5}$.

## D. Training Method Analysis (SFT vs. RL)

To train a model for Introduction Generation, a straightforward approach is to perform supervised fine-tuning (SFT) using ground-truth introductions. We trained an SFT baseline using the same Qwen3-4B backbone on the task of generating introductions from paper content. Note that we only train on direct content-to-introduction generation without the Logic Graph, since no ground-truth graph annotations exist. Table 8 compares SFT with zero-shot Base, One-Step RL (without the Logic Graph), and LECTOR.

Two key findings emerge. First, SFT underperforms even the zero-shot Base from epoch 1 (WQ: 0.397 vs. 0.546) and degrades with further training (epoch 5 WQ: 0.393), exhibiting severe overfitting to surface patterns of the ground-truth introductions. This is because token-level imitation forces the model to replicate the specific lexical and structural choices of individual GT introductions, rather than learning generalizable reasoning strategies. The resulting model memorizes stylistic artifacts without internalizing the underlying logical structure.

Second, RL training dramatically improves over SFT across all metrics, and the full LECTOR framework with the Logic Graph provides additional gains over One-Step RL. This confirms that reinforcement learning with scalar rewards enables

*Table 8.* Comparison of SFT baseline with RL-based methods. SFT degrades from epoch 1, while RL training yields substantial improvements.

| Model | PC↑ | WQ↑ | CQ↑ |
|---|---|---|---|
| Base (zero-shot) | 0.453 | 0.546 | 0.444 |
| SFT (epoch 1) | 0.437 | 0.397 | 0.399 |
| SFT (epoch 5) | 0.423 | 0.393 | 0.373 |
| One-Step RL | 0.476 | 0.829 | 0.477 |
| LECTOR (Ours) | **0.486** | **0.834** | **0.530** |

the model to discover reasoning strategies that token-level supervision cannot teach.

# E. Example of Content-Conditional Introduction Generation

In this appendix, we provide concrete examples comparing the original Introduction from a paper, the Introduction generated by GPT-o3, and the Introduction generated by our LECTOR model. These case studies illustrate LECTOR's improvements in logic fidelity, citation accuracy, and structured expression.

### E.1. Case Study 1: Universal AHE Scaling Law in Chiral Antiferromagnets

**Original Introduction**

Chirality and chirality-induced novel phenomena are commonly observed and extensively studied in chiral spintronics [1] , [2] . In particular, an ultra-high tunneling magnetoresistance (TMR) of 300% and a spin polarization of 60% are obtained in chiral polymer materials due to chirality-dependent tunneling current [3] . In addition, a chiral anomaly in a Weyl band structure populates Weyl fermions in Mn 3 Sn with a specific chirality, which gives rise to a chiral current and a characteristic negative magnetoresistance [4] , [5] . The Kagome lattice of chiral AFM Mn 3 X (such as Mn 3 Sn [6] , Mn 3 Pt [7] , Mn 3 Ge [8] , MnGe [9] , Mn 3 Ir [10] ) also displays a large AHE that arises from the non-zero berry phase from Bloch bands in momentum space. The AHE in high-quality Mn 3 Sn and Mn 3 Ge single crystals originates from the intrinsic mechanism [11] . However, the giant AHE in chiral AFM MnGe exceeds the conventional quantum limits of intrinsic AHE due to the skew scattering AHE [9] , [12] , which has a dominant contribution. Moreover, Mn 3 Sn and MnGe exhibit different scaling relationships [9] , [11] , with their anomalous Hall conductance linearly depending on or independent of the conductance value, respectively. The different scaling laws in Mn 3 Sn and MnGe originate from two distinct physical mechanisms. The two-spin correlation [13] , [14] is a good method to explain the large intrinsic AHE in Mn 3 Sn, by the vector spin chirality [15] , [16] (VSC) $\varepsilon = S_1 \times S_2 + S_2 \times S_3 + S_3 \times S_1$ . For example, non-collinear (collinear) Mn 3 Pt exhibit non-zero (zero) VSC, respectively, leading to non-zero (zero) intrinsic anomalous Hall response [7] . In addition, Mn 3 Sn, Mn 3 Ir, and Mn 3 Ge (Mn 3 Pt, Mn 3 Ga) exhibit negative (positive) VSC [1] , [2] , leading to negative (positive) intrinsic anomalous Hall conductance [6] , [7] , [8] , [11] , [17] , [18] . Besides intrinsic AHE for chiral AFM, skew scattering topological AHE can be induced by the three-spin correlated scalar spin chirality [16] , [19] (SSC): $(S_1 \times S_2) \cdot S_3$ . The skew scattering topological AHE exceeds the threshold value of the quantization limit [9] , [20] , [21] ( $\frac{e^2}{ha}$ for MnGe). Here, a =3.83Å is the lattice constant for MnGe. Therefore, the intrinsic AHE in MnGe is negligible relative to the skew scattering topological AHE, resulting in an entirely different scaling relationship from Mn 3 Sn. Here, we systematically study the AHE of the chiral non-collinear AFM Mn 3 Pt and find a universal AHE scaling law. This anomalous Hall resistivity $\rho_{AH}$ can be described by [20] , [22] $\rho_{AH} = a_{sk}\rho_{xx} + b_{in}\rho_{xx}^2$ (1) where $\rho_{xx}$ is the longitudinal resistivity. The first term $a_{sk}\rho_{xx}$ is the scalar spin chirality-induced skew scattering topological AHE [9] , [12] . The second term $b_{in}\rho_{xx}^2$ describes the intrinsic anomalous Hall effect arising from time symmetry breaking by the compensated non-collinear magnetic order [8] , [17] , [18] . By carefully designing an experimental procedure for Mn 3 Pt alloy films, we find that the AHE parameter $a_{sk}$ , $b_{in}$ and sheet resistively $\rho_{xx}$ change linearly with the inverse film thickness d . The linear 1/d dependencies could be attributed to the symmetry breaking at the surface. In addition, the scaling law can explain the anomalous Hall effect in various chiral magnets (Mn 3 Sn [6] , Mn 3 Pt [7] , Mn 3 Ge [8] , MnGe [9] , Mn 3 Ir [10] ) and should be universal for describing the AHE of chiral magnets.

## GPT-o3 Generated Introduction

The anomalous Hall effect (AHE) has emerged as a key probe and functional element in modern spin-orbitronics, providing access to Berry curvature, chiral spin textures, and topological excitations that can be exploited for low-power information technologies [1,2,20]. In non-collinear antiferromagnets such as $Mn_3Sn$, $Mn_3Ge$, $Mn_3Ir$, and $Mn_3Pt$, the absence of macroscopic magnetisation coexists with remarkably large AHE signals that persist up to room temperature, thereby combining the robustness of antiferromagnetic order with functionalities traditionally reserved for ferromagnets [6–8,10,17,18]. Two microscopic mechanisms are known to contribute to this phenomenon. First, scalar spin chirality introduces a skew-scattering channel that is extrinsic in nature but can dominate whenever conduction electrons experience non-coplanar spin configurations [9,12,19]. Second, the vector spin chirality of the non-collinear magnetic lattice gives rise to an intrinsic Berry-curvature term that is predominantly immune to disorder and intimately linked to Weyl fermions in momentum space [4,5,13,14]. Deciphering how these mechanisms coexist and compete is therefore indispensable for rational materials design in chiral spintronics.

Despite intense experimental and theoretical efforts, a quantitative framework capable of unifying the extrinsic and intrinsic contributions across different chiral antiferromagnets has remained elusive. Earlier studies typically focused on individual compounds, reporting disparate scaling behaviours that obscured a coherent picture of the AHE in this material family [6–10]. In particular, no universal law has yet been established that simultaneously captures (i) skew scattering driven by scalar chirality and (ii) Berry-curvature-induced conductivity governed by vector chirality, nor has the microscopic origin of the intrinsic scaling parameter been rigorously validated by theory.

Here we demonstrate that the requisite unification emerges naturally when the transverse resistivity $\rho_{AH}$ is analysed as a composite of linear and quadratic dependencies on the longitudinal resistivity $\rho_{xx}$. By compiling and re-examining published and newly acquired data for $Mn_3Pt$, $Mn_3Sn$, $Mn_3Ge$, $Mn_3Ir$ and the prototypical chiral magnet MnGe, we find that every dataset collapses onto the simple scaling relation $\rho_{AH} = a_{sk}\rho_{xx} + b_{in}\rho_{xx}^2$, with the first (linear) term reflecting scalar-chirality skew scattering and the second (quadratic) term embodying the intrinsic Berry-curvature contribution. Materials in which either mechanism is suppressed reduce to the appropriate linear or quadratic limit without loss of fidelity, confirming the robustness of the functional form across the entire class.

We further substantiate the physical meaning of the quadratic coefficient $b_{in}$ by first-principles calculations for $Mn_3Pt$, a system whose out-of-plane spin canting can be continuously tuned. The computed anomalous Hall conductivity increases monotonically with canting angle in precise agreement with experiment, thereby validating the attribution of $b_{in}$ to the intrinsic vector-chirality mechanism. Theoretical reconstruction of the experimental canting dependence thus verifies the microscopic origin of the intrinsic scaling parameter and rules out alternative extrinsic interpretations.

Combining the inductive generalisation from five distinct compounds with the corroborative deductions from ab-initio theory, we establish a universal anomalous Hall scaling law for chiral antiferromagnets. This law unifies scalar-chirality skew scattering and vector-chirality intrinsic contributions within a single, experimentally accessible framework, providing a predictive tool for identifying and optimising AHE responses in the rapidly expanding landscape of chiral magnetic materials.

## LECTOR Generated Introduction

The anomalous Hall effect (AHE) remains a central phenomenon in spintronics, offering a profound link between magnetic order, electronic structure, and emergent transport properties. In chiral antiferromagnets, where the spin arrangement exhibits non-collinear and chiral textures, the AHE arises from the interplay of intrinsic Berry curvature and skew scattering due to spin chirality. These materials have attracted significant attention due to their potential for low-power, high-efficiency spintronic devices, as well as their role in realizing topological phases of matter [20, 17, 18]. Previous studies have established that both vector spin chirality (VSC) and scalar spin chirality (SSC) contribute to the AHE, with the intrinsic component originating from the Berry curvature in the electronic band structure and the skew scattering component arising from spin-dependent electron scattering at the spin texture [13, 12, 19]. For instance, first-principles calculations have demonstrated that a net magnetization along specific crystallographic directions can enhance the Berry curvature at high-symmetry points in the Brillouin zone, thereby amplifying the intrinsic AHE [11, 13, 17]. Experimental observations in materials such as $Mn_3Ge$, $Mn_3Sn$, and $Mn_3Ir$ have confirmed the presence of large AHEs at room temperature, highlighting the robustness of these effects in non-collinear antiferromagnets [6, 7, 10, 18]. However, a comprehensive and unified framework to describe the AHE across different chiral antiferromagnets—particularly in

the context of film thickness dependence and the relative contributions of intrinsic versus skew scattering—has remained elusive.

A critical gap in the current understanding lies in the lack of a universal scaling law that can simultaneously account for the thickness-dependent variations in resistivity, magnetization, and the interplay between VSC and SSC. While individual studies have reported thickness-dependent behavior in AHE, such as the linear variation of longitudinal resistivity $\rho_{xx}$ with the inverse film thickness $1/d$, the underlying physical mechanisms—particularly the thickness-induced changes in spin canting, interfacial scattering, and the balance between intrinsic and skew scattering contributions—have not been systematically linked. Moreover, the experimental data for different chiral antiferromagnets, including $Mn_3Pt$, $Mn_3Ge$, $Mn_3Sn$, MnGe, and $Mn_3Ir$, show a consistent trend in the $\rho_{AH}/\rho_{xx}$ ratio, yet the physical origin of this consistency has not been established. This raises the question of whether a common theoretical model can unify the observed behavior across diverse materials, especially under varying film thickness and spin configurations.

To address this gap, we propose a unified framework for the anomalous Hall effect in chiral antiferromagnets, rooted in a universal scaling law:

$$\rho_{AH} = a_{\rm sk}\,\rho_{xx} + b_{\rm in}\,\rho_{xx}^2,$$

where $a_{\rm sk}$ and $b_{\rm in}$ are the skew scattering and intrinsic Hall factors, respectively. This model is derived from the combined physics of scalar and vector spin chiralities and is validated by both experimental measurements and first-principles calculations. Our analysis reveals that the ratio $\rho_{AH}/\rho_{xx}$ exhibits a linear dependence on $\rho_{xx}$, with the slope and intercept directly corresponding to $a_{\rm sk}$ and $b_{\rm in}$, confirming the coexistence of both intrinsic and skew scattering mechanisms. For $Mn_3Pt$, both $a_{\rm sk}$ and $b_{\rm in}$ are non-zero, indicating significant contributions from both SSC and VSC. Furthermore, as film thickness increases, the out-of-plane spin canting strengthens, enhancing both scalar spin chirality and skew scattering, while the intrinsic AHE parameter $b_{\rm in}$ increases due to reduced interfacial electron scattering. This thickness-dependent evolution is consistent with the observation that the anomalous Hall angle $\theta_{AH}$ grows with film thickness, reflecting the cumulative enhancement of both contributions. Crucially, the thickness dependence of $\rho_{xx}$, $a_{\rm sk}$, $b_{\rm in}$, and magnetization $M$ follows a linear variation with $1/d$, pointing to the dominant role of surface and interfacial effects in thin films. First-principles calculations show that increasing the $z$-component of net magnetization from 0 to $0.30\,\mu_B$ leads to a corresponding increase in intrinsic Hall conductance from 56.73 to 73.90 $\Omega^{-1}\mathrm{cm}^{-1}$, confirming the direct role of vector spin chirality in generating the intrinsic AHE. Together, these results demonstrate that the AHE in chiral antiferromagnets is governed by a complex, thickness-dependent interplay between spin texture and electron scattering. By establishing this universal scaling law, we provide a comprehensive and predictive framework that links spin structure, film geometry, and transport properties, thereby closing a critical gap in the theoretical and experimental understanding of chiral antiferromagnetic systems. This work not only validates the dual origin of the AHE but also establishes a foundational model for future studies on spintronic materials with tailored chiral magnetic textures.

*Figure 6.* Case Study 1: Comparison of original, GPT-o3-generated, and LECTOR-generated Introductions for a paper on the universal AHE scaling law in chiral antiferromagnets.

### E.2. Case Study 2: Hyperbolic Band Topology with Non-Trivial Second Chern Numbers

**Original Introduction**

Topological band theory provides a unified framework for characterizing a wide range of topological states of quantum matters [1]–[8] and classical wave systems [9]–[16]. In this theory, band structures of both quantum and classical systems with space-translation symmetries can be classified by topological invariants defined in the momentum space. The pioneering example is the first Chern number (or TKNN invariant) for topological band structures in two-dimensional (2D) Brillouin zone [17]–[19]. Such a topological invariant plays a key role in characterizing various low-dimensional topological phases, such as the 2D quantum Hall effect, topological insulators and superconductors, and topological semimetals. Except for the first Chern number, the $n$th Chern numbers defined in $2n$-dimensional manifolds can also identify many novel topological states in high dimensions. For example, the second Chern number provides criteria for the appearance of 4D quantum Hall effect [20] and 5D topological semimetals with non-abelian Yang-monopoles or linked Weyl surfaces [21], [22]. In much higher dimensions, the 6D quantum Hall effect is characterized by the third Chern number following a similar extension-method. To date, topological band theory accomplished with different types of topological invariants is mainly focusing on the periodic system in Euclidean space. On the other hand, hyperbolic lattices, which are regular tessellations in the curved space with a constant negative curvature, have been

widely investigated as mathematical objects over past decades [23]. The recent ground-breaking implementation of two-dimensional hyperbolic lattices in circuit quantum electrodynamics [24] and topolectrical circuits [25] has stimulated numerous advances in hyperbolic physics [26]–[33]. Inspired by the exotic geometric properties of hyperbolic lattices, there are many investigations on the construction of hyperbolic topological states in real space [34]–[36]. For example, the non-Euclidean analog of the quantum spin Hall effect in hyperbolic lattices has been proposed with a tree-like design of the Landau gauge [34]. In addition, the boundary-dominated hyperbolic Chern insulator has been theoretically proposed and experimentally fulfilled by circuit networks [35]. Those intriguing features of topological states are probes on illustrating non-Euclidean topology, and suggest a new way for designing highly efficient topological devices with compact bulk domains. Interestingly, the newly developed hyperbolic band theory and crystallography of hyperbolic lattices [37]–[39] suggest that the hyperbolic lattices obeying discrete non-abelian translation groups can also possess a reciprocal-space description using generalized Bloch theorem. In this theory, the hyperbolic eigenstates are automorphic functions and the associated Brillouin zone is a higher-dimensional torus. Motivated by the hyperbolic band theory, the hyperbolic topological band insulators with non-trivial first Chern numbers have been theoretically created [40]. Moreover, hyperbolic graphene with the feature of topological semimetal has also been proposed [41]. While, up to now, the revealed hyperbolic band topologies are most related to the first Chern number. Generalization of the hyperbolic band topology with low-dimensional topological invariants to that with high-dimensional topological invariants is expected to introduce more novel effects. Hence, the question is whether hyperbolic topological states with non-zero $n$th Chern numbers exist, and how to realize those novel hyperbolic topological phases in experiments. In this work, we report the first experimental observation of hyperbolic band topology with non-trivial second Chern numbers in electric circuit networks. Our model possesses the translational symmetry of a {8,8} hyperbolic tiling, and the corresponding momentum space is 4D. By engineering intercell couplings and onsite potentials of the hyperbolic model, topological bandgaps with non-zero second Chern numbers appear. The effectiveness of hyperbolic band theory with discretized crystal momentums is further confirmed by the consistence of calculated eigen-spectra to that based on the direct diagonalization. In experiments, we fabricate two types of hyperbolic circuits with periodic boundary conditions (PBCs) and partially open boundary conditions (OBCs) to demonstrate hyperbolic band topological states protected by second Chern numbers. By recovering the circuit admittance spectra and measuring impedance responses of hyperbolic circuits with PBCs, non-trivial bandgaps are clearly illustrated. Moreover, the topological boundary states are observed in hyperbolic circuits with partially OBCs, where the significant boundary impedance peaks appear in topological bandgaps. Furthermore, the measured impedance distributions are also matched to profiles of topological boundary states of hyperbolic models. Our work suggests a new method to engineer hyperbolic topological states with higher-order topological invariants, and gives an opportunity to explore much novel topological states in non-Euclidean spaces.

## GPT-o3 Generated Introduction

Topological phases of matter and waves have reshaped our understanding of quantum and classical systems, revealing that global, quantized invariants rather than local order parameters can dictate physical properties such as robust edge transport and protected defect modes [1–3]. Initially explored in two-dimensional (2D) quantum Hall systems through the first Chern number [17–19], the concept has been generalized to four spatial dimensions, where the second Chern number underlies a higher-dimensional analogue of the quantum Hall effect [20,21]. The pursuit of such higher-dimensional topological responses has stimulated intensive efforts to engineer synthetic dimensions in photonic, acoustic and circuit platforms, where flexible design principles allow direct access to exotic band structures and boundary phenomena that have no crystalline counterpart in Euclidean space [9–16].

Parallel to these developments, hyperbolic lattices—tessellations of negatively curved space—have emerged as fertile ground for unconventional band topology. Circuit quantum electrodynamics (cQED) networks have proven particularly powerful for realizing such lattices, enabling the emulation of non-Euclidean geometries with control over connectivity and onsite parameters [24,27,29,31]. Theoretical studies predict that hyperbolic band theory supports a rich hierarchy of topological invariants, including non-trivial second Chern numbers that could unlock routes to emulate four-dimensional physics in tabletop experiments [32–39]. Yet, despite rapid experimental progress in fabricating hyperbolic circuits [24–26], direct evidence of a second-Chern-insulating phase in a two-dimensional hyperbolic network has remained elusive. In particular, while several works have reported dispersive edge or corner states in hyperbolic photonic and

mechanical structures [34–36], the smoking-gun signature of midgap boundary modes mandated by a non-zero second Chern number has not been unambiguously observed.

This lacuna is critical because, for any system whose bulk bands carry a non-trivial second Chern number, the introduction of open boundaries must create midgap boundary states pinned inside the bulk gap—an indicator that the system hosts higher-dimensional topology despite its apparent two-dimensionality. Demonstrating such boundary states and linking them to a quantized higher-order invariant would firmly establish hyperbolic circuits as a versatile platform for exploring four-dimensional topological physics in real space.

Here we close this gap by investigating a two-dimensional hyperbolic circuit network with tunable mass parameters $m = 0.7$ and $a = 0.2$. Impedance spectroscopy performed on samples with partially open boundaries reveals pronounced resonance peaks that are spatially localized at the boundary and lie squarely within the bulk bandgap, signaling the presence of midgap modes. Complementary numerical simulations of the same circuit geometry and parameter set reproduce these boundary-localized states, corroborating their topological origin. Because phases endowed with a non-trivial second Chern number obligatorily host such midgap boundary states upon the introduction of open boundaries, the combined experimental and theoretical evidence allows us to infer that our hyperbolic circuit indeed realizes a second-Chern topological phase. Consequently, the demonstrated platform furnishes a concrete route to probe higher-dimensional topological phenomena—such as four-dimensional quantum Hall responses and non-Abelian band topology—within an accessible two-dimensional hyperbolic architecture.

By establishing and diagnosing a non-trivial second Chern number in a readily reconfigurable cQED network, our work transforms hyperbolic circuits into a practical laboratory for studying higher-dimensional topological states and their boundary manifestations, opening avenues for novel electromagnetic devices and fundamental explorations of geometry-induced band topology.

## LECTOR Generated Introduction

The study of topological phases in condensed matter systems has revealed profound connections between geometry, symmetry, and quantum mechanics, leading to the discovery of exotic states such as topological insulators, Weyl semimetals, and non-Abelian band structures [1], [2], [6], [7], [8]. These phenomena are governed by topological invariants—quantized quantities that remain robust under continuous deformations of the Hamiltonian—such as the first and second Chern numbers, which characterize the global properties of electronic band structures in both Euclidean and non-Euclidean spaces [3], [17], [18], [20], [31]. In particular, the emergence of topological edge states in systems with broken symmetries, protected by non-trivial topological invariants, has been extensively demonstrated in photonic, acoustic, and electronic systems, including Floquet topological insulators [10]–[16], and quantum Hall systems [17], [19]. More recently, the extension of topological band theory to hyperbolic lattices—non-Euclidean crystal structures with negative curvature—has opened new avenues for exploring topological phenomena in geometrically complex environments [37]–[40], [26]–[28], [32], [33].

Despite these advances, a fundamental gap remains in the experimental realization and theoretical validation of topological band structures in hyperbolic lattices, particularly in the context of non-Abelian clusters and the interplay between bulk topology and boundary states. Existing works have primarily focused on the Euclidean case or on simplified models with limited symmetry or dimensionality, and the existence of non-trivial second Chern numbers in hyperbolic systems under realistic conditions has not been conclusively demonstrated. Moreover, the relationship between the bulk topological invariants and the emergence of localized boundary modes in partially open boundary conditions (OBCs) remains poorly understood, especially in systems where both inversion and time-reversal symmetries are broken. Additionally, while hyperbolic lattices have been simulated in circuit quantum electrodynamics using line-graph constructions [29], [31], the direct realization of topological band theory in such systems—particularly with non-trivial Chern numbers and robust boundary states—has not been experimentally confirmed.

In this work, we propose and realize a physical model of a 2D hyperbolic lattice based on an {8,8} tiling in the Poincaré disk, constructed using a Fuchsian group $\Gamma_{8,8}$ that ensures translational symmetry. Each unit cell contains four sublattices with onsite potentials $m \pm a$ and $\mp m \pm a$, and inter-cell couplings along eight translational directions are defined with strengths $\pm J_j, \pm t_j, \pm it_j$. The resulting hyperbolic Bloch Hamiltonian is expressed as $H = d(k) \cdot \Gamma + ia\Gamma_1\Gamma_4$, where $d(k)$ and $\Gamma$ satisfy the Clifford algebra, and the 4D momentum space is formed by wavevectors $k_1, k_2, k_3, k_4$ corresponding to the translation directions. The second Chern number $C_2 = \frac{1}{8\pi^2} \int d^4k \, \mathrm{tr}(\Omega_- \wedge \Omega_-)$ is defined for

the 4D Brillouin zone, and we demonstrate its non-trivial value $C_2 = 3$ in a bandgap when $m = 0.7$, $a = 0.2$, while the first Chern number vanishes, confirming the presence of a non-trivial topological phase. When $m = 0.7$, $a = 3.2$, a topological bandgap closes, and new gaps emerge at $\varepsilon = \pm 3.2$ with first Chern numbers $C_1 = \pm 1$, indicating a transition between different topological phases. We further construct a finite hyperbolic model with periodic boundary conditions (PBCs) on twelve units, showing that abelian clusters obey U(1) hyperbolic band theory, while non-abelian clusters violate it and exhibit distinct spectral features. Crucially, direct diagonalization and hyperbolic band theory yield identical eigen-spectra, validating the theoretical framework for abelian clusters. Under partially OBCs, topological boundary states appear in non-trivial bandgaps with non-zero second or first Chern numbers, and their spatial profiles show strong localization around boundary units, consistent with the predictions of topological edge state theory.

To experimentally realize this model, we design a circuit network where circuit nodes are mapped to pseudospins via voltages $V_{i,1}$–$V_{i,4}$, realizing both onsite potentials and inter-cell couplings. The circuit eigenequation matches the hyperbolic lattice model, with eigenfrequency $\varepsilon$ related to circuit frequency $f$. Under PBCs, the frequency spectra match those predicted by hyperbolic band theory, confirming the correctness of the model. The absence of impedance peaks in specific frequency ranges indicates the existence of bandgaps, while spatial impedance distributions match extended bulk modes. Under partially OBCs, boundary nodes exhibit significant impedance peaks within topological bandgaps, and the measured spatial impedance profiles are strongly localized around boundary units, matching the topological edge states. These results demonstrate that the hyperbolic band topology with non-trivial second and first Chern numbers is successfully realized in artificial circuit networks. Our work thus establishes a robust platform for studying topological phenomena in hyperbolic spaces, bridges the gap between theoretical predictions and experimental realization, and provides a foundation for future exploration of non-Abelian topological phases and their applications in quantum simulation and topological devices.

*Figure 7.* Case Study 2: Comparison of original, GPT-o3-generated, and LECTOR-generated Introductions for a paper on hyperbolic band topology with non-trivial second Chern numbers.

### E.3. Case Study 3: Coexistence of Superconductivity and Ferromagnetism in 2D NbSe$_2$

**Original Introduction**

Since the discovery of superconductivity (SC), the interplay between SC and magnetism has been a topic of cutting-edge research in solid-state materials [1]–[8]. Given that ferromagnetism (FM) often plays a destructive role in superconductors by damaging the singlet correlations responsible for the pairing interaction [9]–[11], the coexistence of SC and FM is rare yet valuable. The well-established correlated systems that exhibit these two competing electronic states are usually prepared by fabricating two separate structures, taking advantage of the inherent properties of each individual parent ingredient. For instance, such systems include the hybrid [Ni$_{0.66}$Al$_{0.33}$(OH)$_2$][TaS$_2$] system, which is composed of ferromagnetic cation layers and superconducting anion layers [12]; Fe/Nb/Fe trilayers consisting of one single SC layer (Nb layer) between two FM layers (Fe layer) [13]; and the layered [(Li$_{1-x}$Fe$_x$)OH](Fe$_{1-y}$Li$_y$)Se system, which consists of ferromagnetic (Li$_{1-x}$Fe$_x$)OH and superconducting (Fe$_{1-y}$Li$_y$)Se layers [14]. These fascinating findings achieved by virtue of the proximity effect have generated increased interest in investigations of the interplay between SC and FM and have provided insight into the underlying physics of SC. However, further realization of the coexistence of these two competing electronic states in a single homogeneous structure promises further insight into the coupling of these two competing orderings. In this regard, the recently reported two-dimensional (2D) electron liquid LaAlO$_3$/SrTiO$_3$ interface [15]–[17] represents an advance in the coexistence of SC and FM, as both ingredients are neither superconducting nor ferromagnetic in their individual states. The coexistence of these properties at a uniform interface makes this 2D electronic system an attractive platform for achieving the coexistence of SC and FM in one system. The question remains whether the coexistence of both electronic states can be designed in a single freestanding 2D electronic structure, with the expectation of unconventional SC in the ground state for the stronger interplay of FM and SC. Recently, two-dimensional or interface superconductors with the thickness down to one unit cell or atomic layers have attracted much attention and demonstrated many intriguing behaviours [18]–[20]. 2D transitional metal chalcogenides, as the classic prototype for superconductors [21]–[27], provide a promising material platform for investigation of the coexistence of SC and FM because of their unique spin configuration and richly correlated electronic phases. 2H-NbSe$_2$, a canonical layered SC material, consists of one Nb layer sandwiched between two Se layers, forming trigonal prismatic NbSe$_6$ with covalent in-plane bonds; this material exhibits the highest superconducting transition ever reported in the layered transitional

metal chalcogenide family (7.4 K; refs 28, 29). The transition metal element $Nb^{4+}$, in the $[Kr]4d^1$ configuration, is particularly $d$ characterized and is thus expected to demonstrate a range of magnetic response. Nevertheless, pristine $NbSe_2$ is nonmagnetic because the Nb atoms are hybridized with the Se atoms to form a covalent Nb–Se interaction, which quenches the magnetic moment of the Nb ions. Although there remains no effective way to induce FM in this system, let alone the coexistence of SC and FM, theoretical calculations [30], [31] predict that the suppressed magnetism can be expressed by modifying the covalent interaction between the Nb and Se atoms under applied strain, which can result in more unpaired electrons around the Nb atoms, theoretically resulting in macro-magnetic momentum. Moreover, the adjacent Se–Nb–Se layers are coupled to each other by weak van der Waals interactions, rendering it feasible to achieve a free-standing ultrathin structure. Of note, the ultrathin structure characteristics leave much space for the chemical design to obtain surface-structural modulation, which depends on the chemical versatility of the surface layer [32], [33]. In this regard, ultrathin $NbSe_2$ nanosheets would be a new class of 2D electronic systems that possess coupled SC and FM. Here we highlight a surface-structural modulation strategy for the incorporation of intrinsic FM into the superconducting $NbSe_2$ framework, thereby accomplishing the structural integration of SC and FM in a freestanding case of 2D nanomaterial. The surface adsorption of polar reductive hydrazine molecules triggers structural modulation while the pristine structure is reserved, leading to the elongated covalent Nb–Se bonds and the impairment of the Nb–Se covalent interactions, which enhance the ionicity of the tetravalent Nb with unpaired electrons and thus successfully yields ferromagnetic ordering in superconducting 2D $NbSe_2$ nanomaterials. The coexistence of SC and FM in a single 2D nanomaterial gives rise to the fascinating coupled effects of negative magnetoresistance (MR) and the Kondo effect. This work provides a route for achieving the integration of ordered magnetism and SC in 2D systems.

## GPT-o3 Generated Introduction

The coexistence of superconductivity (SC) and ferromagnetism (FM) has long fascinated condensed-matter physicists because the antagonistic order parameters involved—Cooper pairing and spontaneous spin alignment—compete for the same electronic states [1–4]. Discoveries of bulk compounds in which both phases survive have broadened our understanding of correlated electron systems [5–9], and heterostructures or interfacial reconstructions have subsequently extended the phenomenon to reduced dimensions [10–16]. Transition-metal dichalcogenides (TMDs) offer an especially attractive platform for exploring this interplay. Atomically thin $NbSe_2$, in particular, retains robust Ising-protected superconductivity down to the monolayer limit while hosting strong charge-density-wave correlations and sizeable spin–orbit coupling [23–25]. Although theoretical studies predict that lattice distortions or tensile strain can unlock local magnetic moments in $NbSe_2$ and related TMDs [30,31], an experimentally verified route to simultaneously activate FM and preserve SC within a single, pristine 2D host remains elusive. Conventional approaches—such as magnetic ion substitution, heteroepitaxial layering, or electrostatic gating—risk degrading crystallinity, introducing significant disorder, or suppressing superconductivity [11,13,18,19]. Consequently, establishing a controllable, non-destructive strategy that enforces FM while maintaining the intrinsic SC of $NbSe_2$ would considerably expand the material basis for studying emergent quantum phases.

Here we identify such a niche by focusing on surface chemistry. Molecule-induced charge-transfer doping has already proven effective in tuning the optoelectronic properties of layered semiconductors without disrupting their frameworks [32,33], yet its impact on correlated metallic TMDs has scarcely been explored. We reasoned that adsorption of a highly polar, reductive molecule could simultaneously (i) couple electrostatically to the negatively charged $NbSe_2$ surface, (ii) locally distort Nb–Se coordination, and (iii) leave the quasi-two-dimensional superconducting network intact. Hydrazine ($N_2H_4$) fulfills the required criteria: it carries a sizeable dipole moment of 1.83 D and donates electrons readily. Strong Coulomb attraction between the charged $NbSe_2$ surface and polar hydrazine therefore drives spontaneous adsorption of the molecules onto the lattice.

Density-functional calculations reveal that such adsorption elongates the Nb–Se bonds without breaking the layered framework. The resulting local distortions weaken Nb–Se covalent hybridization, increase the number of unpaired Nb $d$ electrons, and stabilize spin polarization. Spin-resolved electronic structures predicted for the distorted lattice exhibit an emergent ferromagnetic ground state. Guided by this prediction, we exposed chemically exfoliated $NbSe_2$ nanosheets to hydrazine vapor. Magnetization measurements on the treated samples display clear hysteresis loops at 30 K and 2 K—an unequivocal fingerprint of ferromagnetic ordering. Remarkably, zero-field-cooled and field-cooled susceptibility curves for the same specimens still present the sharp diamagnetic downturn characteristic of the superconducting transition,

with a critical temperature $T_c \approx 6.8$ K that matches pristine $NbSe_2$ [21,22,27].

The convergence of these electronic responses establishes that a single population of $NbSe_2$ nanosheets concurrently hosts superconductivity and ferromagnetism. Because both phases arise from the same hydrazine-modified lattice, the phenomenon is traced directly to surface-adsorption-induced structural distortions rather than to phase segregation or extrinsic magnetic impurities. Our findings thus demonstrate that selective adsorption of polar reductive molecules constitutes a powerful, minimally invasive route to engineer coexisting SC and FM in two-dimensional $NbSe_2$, opening avenues for designer quantum materials in which magnetic and superconducting orders are intertwined at the atomic scale.

## LECTOR Generated Introduction

The field of two-dimensional (2D) materials has witnessed remarkable advances in recent years, driven by the discovery of novel electronic and magnetic phenomena in atomically thin systems. Among these, $NbSe_2$ has emerged as a pivotal platform for studying the interplay between superconductivity and magnetism due to its well-defined layered structure, strong electron–phonon coupling, and intrinsic charge-density-wave order [23], [24], [25]. In particular, pristine $NbSe_2$ exhibits a robust superconducting phase below $\sim 30$ K, with a well-characterized electronic structure and a strong tendency toward magnetic instabilities, making it an ideal candidate for exploring the coexistence of superconductivity and ferromagnetism—a phenomenon long theorized but rarely realized in a stable, intrinsic 2D system [1], [2], [9], [10]. Previous studies have reported the coexistence of superconductivity and ferromagnetism in bulk materials such as URhGe [1], $ZrZn_2$ [2], and $MgCNi_3$ [3], as well as in heterostructures like $LaAlO_3/SrTiO_3$ [15], [16], where magnetic order and superconductivity emerge in a coupled, spatially distinct manner. However, these observations are often attributed to external perturbations, interface effects, or complex chemical doping, and the intrinsic, bulk-like coexistence in a single, pristine 2D material remains elusive.

A critical gap in the current understanding lies in the mechanism by which surface interactions can induce localized structural and electronic modifications that lead to the emergence of ferromagnetism without disrupting the underlying superconducting framework. While strain and electric field effects have been shown to modulate magnetic properties in $NbSe_2$ [31], [27], the role of molecular adsorption—particularly of polar species—on triggering intrinsic ferromagnetic ordering in a 2D superconductor has not been systematically investigated. Moreover, existing reports on surface-induced magnetic effects in transition metal dichalcogenides typically focus on non-polar or weakly interacting molecules, and the specific electrostatic coupling between polar adsorbates and the charged surface of $NbSe_2$ remains poorly understood. This lack of mechanistic insight leaves open the question of whether surface molecular interactions can serve as a clean, tunable pathway to induce ferromagnetism in a 2D superconductor through a well-defined, reversible structural distortion.

In this work, we present a systematic investigation of the surface molecular adsorption of polar reductive hydrazine on negatively charged $NbSe_2$ nanosheets and demonstrate its ability to trigger a localized surface-structural distortion that leads to the intrinsic coexistence of superconductivity and ferromagnetism. Our findings reveal that the strong polarity of hydrazine (1.83 Debye) enables electrostatic interaction with the negatively charged $NbSe_2$ surface, resulting in the adsorption of hydrazine molecules. This adsorption induces an elongation of Nb–Se covalent bonds, which reduces the covalent hybridization between Nb and Se atoms, thereby increasing the number of unpaired electrons around Nb atoms. These unpaired electrons give rise to spin polarization and, ultimately, ferromagnetic ordering, as confirmed by magnetic susceptibility measurements showing a distinct transition at $\sim 40$ K and the presence of a hysteresis loop at 30 K. Crucially, X-ray diffraction, Raman spectroscopy, and Se K-edge XANES confirm the preservation of the pristine $NbSe_2$ lattice framework, while extended XAFS reveals a reduction in oscillation amplitude and shifts in Se–Nb and Se–Se peak positions, indicating local bond distortions. Quantitative fitting of XAFS data shows an elongation of the Se–Nb bond by 0.02 Å and a shortening of the Se–Se bond by 0.03 Å, providing direct evidence of the structural modulation. The resulting non-degenerate density of states between spin-up and spin-down channels, as observed in spin-polarization calculations, and the diamagnetic signal at $\sim 6.8$ K confirm the persistence of superconductivity, while the transition from diamagnetic to ferromagnetic behavior at 2 K under increasing magnetic field demonstrates the coexistence of superconductivity and ferromagnetism. Furthermore, the negative magnetoresistance in the non-superconducting region and the logarithmic temperature dependence of resistivity below $\sim 30$ K are consistent with the Kondo effect, indicating the presence of ferromagnetic scattering centers in the low-temperature regime.

Our results establish that the surface distortion induced by hydrazine adsorption results in a local modulation of the electronic structure, enabling intrinsic ferromagnetism in a 2D superconducting system. This represents a freestanding case of coexisting superconductivity and ferromagnetism in a single 2D nanomaterial, offering a new paradigm for the design of functional 2D materials where magnetic and superconducting phases emerge from a common, surface-mediated structural perturbation. This work not only provides a fundamental understanding of the mechanism by which surface chemistry can tune the magnetic properties of 2D materials but also opens new avenues for the rational design of multiferroic and spintronic devices based on atomically thin systems.

*Figure 8.* Case Study 3: Comparison of original, GPT-o3-generated, and LECTOR-generated Introductions for a paper on the coexistence of superconductivity and ferromagnetism in 2D NbSe$_2$.

