# OpenReview forum: "LECTOR: Joint Learning of Scientific Reasoning Graphs and Introduction Generation"
_ICML.cc/2026/Conference — ICML 2026 regular_

### Official Review · Reviewer_iWNk · 2026-03-06

**Soundness:** 3
**Presentation:** 4
**Significance:** 2
**Originality:** 3
**Overall Recommendation:** 4
**Confidence:** 4

**Summary:**

This paper proposes Content-Conditional Introduction Generation (CCIG), a new task in which the model must write a paper’s Introduction from the paper body. To solve this task, the authors introduce a framework that extracts a reasoning logic graph as an intermediate representation and then generates the Introduction from that graph and the citation list. Training is done jointly with RL over the full trajectory from paper body to graph to Introduction.
Empirically, the paper builds a dataset of 10,200 Nature Communications papers and reports that LECTOR substantially improves over its Qwen3-4B backbone and achieves near parity with SOTA Models. Ablations further study the role of graph quality, joint optimization, and individual reward terms.

**Compliance With Llm Reviewing Policy:**

Affirmed.

**Final Justification:**

The rebuttal mostly resolves my concerns with explanations on the dependence on GT Introduction, judge robustness and case studies.

**Key Questions For Authors:**

1. Could the authors provide several qualitative case studies showing the paper body, the extracted reasoning graph, the generated introduction, and the reference introduction?

**Limitations:**

The paper should acknowledge the technical limitations of the method, such as coupling to gold introductions in rewards and evaluation, and uncertainty about cross-domain generalization.

**Strengths And Weaknesses:**

Strengths:
1. The paper tackles a meaningful formulation of scientific writing: generating an Introduction from detailed paper content. The task and methods are well defined.
2. The use of a reasoning graph, inspired by philosophical work, as an explicit intermediate representation is conceptually interesting.
3. The experimental section includes several useful ablations. In particular, Table 3 and Table 4 provide evidence that better graph extraction and joint training both contribute to downstream writing quality.

Weaknesses:
1. Both training rewards and evaluation metrics depend heavily on the ground-truth Introduction. It is hard to tell whether the model is actually reasoning from the evidence or just learning how to copy a specific introduction.
2. The paper uses a larger Qwen3-235B model for reward modeling and judging, while the trained model is Qwen3-4B. The paper does not provide a convincing robustness analysis across alternative judges or judge families, so it is unclear how stable the reported gains are.

---

> ### Author Rebuttal · Authors · 2026-03-31
>
> We thank the reviewer for recognizing the CCIG formulation, the reasoning graph as an explicit intermediate representation, and the useful ablations. We address each concern below.
>
> **W1: Dependence on GT Introduction—Copying vs. Reasoning.** We argue this is unlikely for three reasons:
>
> *(1) RL reward design prevents copying.* Unlike SFT which provides dense token-level supervision, our RL reward uses GT only as a high-level reference for entity coverage and entailment—the model never sees token-by-token GT signals. Our SFT baseline (same Qwen3-4B) confirms this: SFT underperforms even zero-shot Base from epoch 1 (WQ: 0.397 vs. 0.546) and degrades further. It is SFT that exhibits genuine overfitting to GT, not LECTOR.
>
> *(2) Human evaluation breaks the GT dependency.* 8 domain experts scored LECTOR, Base (Qwen3-4B zero-shot), and Original (GT) on 20 test papers across four dimensions (1–5 scale):
>
> | Dimension | Original | Base | LECTOR |
> |---|---|---|---|
> | Logical Coherence | 3.74 | 2.51 | **4.05** |
> | Writing Quality | 3.54 | 2.51 | **3.91** |
> | Citation Integration | **3.21** | 2.06 | 2.99 |
> | Completeness | 3.59 | 2.46 | **3.99** |
> | Overall | 3.52 | 2.38 | **3.73** |
> | Ranked 1st | 25.6% | 0.6% | **73.8%** |
>
> Experts judged quality from their own expertise, not GT similarity. LECTOR was ranked 1st by **73.8%** of 160 expert votes and surpasses GT in 3 of 4 dimensions—results unexplainable by mere copying.
>
> *(3) Qualitative improvements over GT.* Our case studies (see Q1 below) show that LECTOR produces structurally superior introductions compared to GT—with clearer rhetorical organization and more explicit gap articulation—improvements that cannot be explained by copying.
>
> **W2: Judge Robustness.** We address this from two perspectives:
>
> *(1) GQ judge robustness across 4 model families.* We evaluated GQ accuracy using 4 different LLM judges:
>
> | Judge | Approval Rate | Agreement w/ Majority Vote |
> |---|---|---|
> | Gemini-2.5-Pro | 9.9% | 96.2% |
> | Majority Vote (≥3/4) | 11.1% | — |
> | Qwen3-235B (Ours) | 21.6% | 75.9% |
> | GPT-5.4 | 23.4% | 85.7% |
> | Claude-Opus-4-6 | 62.1% | 50.4% |
>
> Judges vary in strictness, but Qwen3-235B occupies a moderate position closest to GPT-5.4 (21.6% vs. 23.4%), with 75.9% agreement with the majority vote. Our evaluation conclusions are not artifacts of a particular judge's bias.
>
> *(2) Human–LLM alignment for WQ/CQ.* Our human evaluation directly validates that LLM judgments align with human experts: Spearman ρ=0.815 (p<0.001) and Krippendorff's α=0.758 (substantial agreement)—stronger evidence than multi-LLM cross-validation, which only proves LLM–LLM consistency.
>
> **Q1: Qualitative Case Studies.** We have included two examples in Appendix C comparing LECTOR-generated introductions with GT. We will expand them into more detailed case studies in the revision. Here we highlight key observations from the chiral antiferromagnet AHE paper:
>
> *(i) Rhetorical structure.* LECTOR generates a clear Swales' CARS structure (Territory→Niche→Contribution), while the GT mixes background, materials, and objectives without clear organization.
>
> *(ii) Logical coherence.* The GT lists facts with flat connectives ("Moreover..., Besides..., In addition...") lacking causal progression. LECTOR instead constructs an explicit logical chain: phenomenon→mechanism classification→existing findings→unresolved problem→proposed solution, with each step grounded in the reasoning graph.
>
> *(iii) Gap and contribution articulation.* LECTOR explicitly states the research gap ("A critical gap in the current understanding lies in...") and the contribution ("closing a critical gap"), which the GT leaves implicit and scattered across technical details.
>
> We will add more case studies from different subdomains in the revision.
>
>
> **Limitations.** We acknowledge the coupling to GT introductions as a valid concern, though our human evaluation and SFT comparison provide evidence against mere copying. Regarding cross-domain generalization, our dataset already spans multiple subdisciplines (pure physics, chemistry, engineering) within Nature Communications. We agree that broader cross-domain validation is important and plan to extend to other disciplines and journal sources in future work.
>
> **Summary.** We have provided: (1) three lines of evidence (reward design analysis, human evaluation, and qualitative cases) demonstrating that LECTOR reasons from evidence rather than copying GT; (2) judge robustness analysis across 4 model families plus human–LLM alignment (ρ=0.815); (3) detailed case studies with traceable graph-to-text mapping; and (4) an honest discussion of limitations with plans for cross-domain extension. We hope these address the reviewer's concerns.

---

> > ### Author Rebuttal · Reviewer_iWNk · 2026-04-03
> >
> > Thanks for the detailed response. The rebuttal mostly resolves my concerns with explanations on the dependence on GT Introduction, judge robustness and case studies. I will raise the score accordingly.

---

> > > ### Author Response · Authors · 2026-04-07
> > >
> > > We are glad that our rebuttal has resolved the concerns, and thank the reviewer for the constructive questions and suggestions throughout the review process. The recognition of our work and the updated score are greatly appreciated.

---

### Official Review · Reviewer_2Wn5 · 2026-03-11

**Soundness:** 2
**Presentation:** 3
**Significance:** 3
**Originality:** 3
**Overall Recommendation:** 3
**Confidence:** 4

**Summary:**

This paper proposes LECTOR, a framework designed to generate scientific paper Introductions conditioned on the main body of the paper (Methods, Results, Analysis, Citations). The research examines an important concept by treating introduction writing as a structured reasoning task rather than a standard text-completion problem. The method extracts a Reasoning Logic Graph as an intermediate step and then uses a joint reinforcement learning approach (simplifed PPO without a separate reward model) to optimize both the graph extraction and the final text generation. Overall, the work analyzes a central aspect of AI-assisted writing: maintaining logical consistency and avoiding citation hallucinations. The authors construct a dataset from Nature Communications and demonstrate that their fine-tuned 4B parameter model outperforms its base version and competes with larger proprietary models on several automated metrics.

**Compliance With Llm Reviewing Policy:**

Affirmed.

**Final Justification:**

Thanks the detailed reponse from the authors. The rebuttal partially resolves my concerns due to the newly added supplementary materials such as human and SFT baseline, I will raise my score finally.

**Key Questions For Authors:**

1. Why is the drop in Graph-Write Alignment (GW) acceptable? If the text diverges from the graph, how can you guarantee the model isn't just hallucinating scientifically plausible but incorrect claims to boost its fluency score?

2. Why is there no human evaluation? LLM judges are known to have severe biases toward their own outputs and specific writing styles. Will you provide human expert evaluations on the logical fidelity of the generated introductions?

3. Can you provide a standard SFT baseline? We need to see if the complex Logic-Expression Co-Reinforcement Learning pipeline actually outperforms a model simply fine-tuned on the ground-truth graphs and introductions.

**Limitations:**

No. The authors discussed paper mills in the impact statement, but totally missed discussing the severe limitations and potential biases of relying exclusively on LLM-as-a-judge for evaluating complex scientific reasoning.

**Strengths And Weaknesses:**

Strengths:
1. Formulating the task as Content-Conditional Introduction Generation (CCIG) is much more realistic than just prompting an LLM with a title and abstract.
2. using a logic graph as an information bottleneck is a solid idea to force the model to ground its claims.
3. The training pipeline is memory-efficient. Dropping the standard reward model in favor of rule-based and LLM-verifier rewards is a smart practical choice.

Weaknesses:
1. The Graph-Write Alignment (GW) metric actually drops after RL training (from 0.682 to 0.623 in Table 2). The authors claims the model learns to prioritize fluency over strict node-to-text translation. But if the generated text doesn't align with the graph, the entire premise of grounding the generation on a verifiable graph is defeated. It looks like the model just learned to hallucinate more fluently to game the Writing Quality (WQ) reward.
2. Lacking of human evaluation. Relying entirely on an LLM-as-a-judge (Qwen3-235B) for metrics like Academic Quality is a big concern for me. The RL policy is highly likely just reward hacking the LLM judge's specific stylistic preferences.
3. The baselines are also weak. The only open-source model tested is the exact same base model (Qwen3-4B). Why not compare against a standard SFT version of the model to prove the RL actually adds value? Comparing your specifically tuned model against zero-shot APIs and an untrained base model isn't a fair fight.
4. Hardcapping the graph size at 50 nodes seems arbitrary.

---

> ### Author Rebuttal · Authors · 2026-03-31
>
> We thank the reviewer for recognizing the CCIG formulation, the logic graph as an information bottleneck, and our memory-efficient training pipeline. We address each concern below.
>
> **W1 & Q1: GW Drop and Hallucination Concern.** We respectfully disagree that the GW decrease implies "hallucinating more fluently." The GW metric measures lexical overlap between graph nodes and generated text. After RL training, the model learns to paraphrase and integrate graph content into fluent academic prose rather than directly transcribing node labels—this naturally reduces lexical overlap while preserving semantic fidelity.
>
> We provide three lines of evidence against hallucination:
>
> *(1) Human evaluation confirms logical fidelity.* 8 domain experts scored LECTOR, Base (Qwen3-4B zero-shot), and Original (GT) on 20 test papers across four dimensions (1–5 scale):
>
> | Dimension | Original | Base | LECTOR |
> |---|---|---|---|
> | Logical Coherence | 3.74 | 2.51 | **4.05** |
> | Writing Quality | 3.54 | 2.51 | **3.91** |
> | Citation Integration | **3.21** | 2.06 | 2.99 |
> | Completeness | 3.59 | 2.46 | **3.99** |
> | Overall | 3.52 | 2.38 | **3.73** |
> | Ranked 1st | 25.6% | 0.6% | **73.8%** |
>
> LECTOR achieves the highest Logical Coherence (4.05) and Completeness (3.99), both directly reflecting whether the generated content is grounded and faithful. If the model were hallucinating, experts would penalize it on these dimensions. Instead, LECTOR surpasses even the GT introductions.
>
> *(2) GW drop accompanies large quality gains.* The GW decrease (Δ=−0.058, dz=−0.36) is small-to-medium in effect size, while WQ increases by +0.289 (dz=1.59) and CQ by +0.086 (dz=0.52)—both with large effect sizes and non-overlapping 95% CIs (full statistical table provided in our response to Reviewer tk7z). This pattern is consistent with the model learning to express graph content more naturally rather than hallucinating: if the model were gaming WQ by ignoring the graph, CQ (which measures citation accuracy against the paper) should also degrade, but it improves significantly.
>
> *(3) Qualitative evidence.* In our case study (Appendix C, chiral antiferromagnet AHE paper), LECTOR generates a clear Swales' CARS structure (Territory→Niche→Contribution) with each claim traceable to specific graph nodes. LECTOR explicitly articulates the research gap ("A critical gap in the current understanding lies in...") and states the contribution ("closing a critical gap"), while the GT leaves these implicit. The generated text paraphrases graph content into coherent academic prose—precisely the behavior that reduces GW while improving actual quality.
>
> **W2 & Q2: Human Evaluation.** We have now conducted a comprehensive human evaluation. As shown in the table above, 8 domain experts independently evaluated 20 randomly sampled test papers. LECTOR was ranked 1st by **73.8%** of 160 expert votes (Spearman ρ=0.815 between human and LLM judgments, Krippendorff's α=0.758). These results directly validate our LLM-as-Judge framework and confirm that LECTOR's improvements reflect genuine quality gains rather than reward hacking.
>
> **W3 & Q3: SFT Baseline.** We have added the requested SFT baseline using the same Qwen3-4B backbone trained on GT introductions:
>
> | Model | PC↑ | WQ↑ | CQ↑ |
> |---|---|---|---|
> | Base (zero-shot) | 0.453 | 0.546 | 0.444 |
> | SFT (epoch 1) | 0.437 | 0.397 | 0.399 |
> | SFT (epoch 5) | 0.423 | 0.393 | 0.373 |
> | One-Step RL | 0.476 | 0.829 | 0.477 |
> | LECTOR (Ours) | **0.486** | **0.834** | **0.530** |
>
> Two key findings: (1) SFT underperforms even zero-shot Base from epoch 1 (WQ: 0.397 vs. 0.546) and degrades monotonically, exhibiting severe overfitting to surface patterns rather than learning reasoning capabilities. (2) RL training dramatically improves over SFT (WQ: 0.397→0.834), and the Logic Graph provides further gains on top of One-Step RL (WQ:0.829→0.834, CQ: 0.477→0.530). This confirms that our RL pipeline and Logic Graph are both essential—not just an over-engineered alternative to SFT.
>
> **W4: 50-Node Graph Cap.** The 50-node limit is not a hard constraint but a soft guidance specified in the prompt. Without an explicit upper bound, the model tends to generate excessively granular graphs with redundant nodes, degrading writing quality. Too few nodes fail to capture sufficient information. We selected 50 as a balanced value covering the vast majority of papers while avoiding degenerate outputs.
>
> **Summary.** We have provided: (1) three lines of evidence (human evaluation, quantitative analysis, and qualitative cases) showing that GW decrease reflects paraphrasing, not hallucination; (2) human evaluation with 8 domain experts (73.8% first-rank, ρ=0.815) validating our automatic metrics; (3) SFT baseline confirming the necessity of RL training and the Logic Graph; and (4) empirical justification for the 50-node cap. We hope these address the reviewer's concerns.

---

> > ### Author Rebuttal · Reviewer_2Wn5 · 2026-04-02
> >
> > Thanks the detailed reponse from the authors. The rebuttal partially resolves my concerns due to the newly added supplementary materials such as human and SFT baseline, I will raise my score finally.

---

> > > ### Author Response · Authors · 2026-04-07
> > >
> > > We sincerely thank the reviewer for the positive re-evaluation and for raising the score. We are encouraged that the human evaluation and SFT baseline have partially addressed your concerns.
> > >
> > > If appropriate and at your convenience, we would greatly appreciate it if you could specify any remaining follow-up questions or unresolved concerns. We are fully committed to providing additional experiments, analyses, or clarifications to further address your questions.
> > >
> > > Thank you again for your constructive engagement with our work.

---

### Official Review · Reviewer_tk7z · 2026-03-12

**Soundness:** 2
**Presentation:** 2
**Significance:** 3
**Originality:** 3
**Overall Recommendation:** 2
**Confidence:** 3

**Summary:**

The authors introduce LECTOR, a Logic-Expression Co-Reinforcement Learning framework, to solve the Content-Conditional Introduction Generation (CCIG) task. LECTOR consists of first constructing a logic-reasoning graph from the paper's main body (methods, results, analyses, and citations), and then using the Logic-Expression Co-Rewarding mechanism to jointly optimize for both the graph’s structural fidelity and the final narrative’s quality. In their experiments, the authors use a dataset of 10,200 Nature Communications papers (10,000/100/100 train/val/test split) and a Qwen3-4B backbone. They find consistent improvements in both logic fidelity and introduction generation quality metrics compared to various out-of-the-box LLMs, including GLM-4.7, OpenAI o3, and Gemini 2.5 Pro.

**Compliance With Llm Reviewing Policy:**

Affirmed.

**Final Justification:**

The authors failed to address my main concern regarding the performance of LECTOR compared to o3.

**Key Questions For Authors:**

- As both the reward and parts of the evaluation use the reference introduction, how do you rule out overfitting?
- Can you provide confidence intervals or significance tests?
- What evaluation criteria would you label as a "true" benchmark vs alignment with the human counterpart?

**Limitations:**

The authors do not fully disclose all limitations of their work. For example, the authors could discuss the fuzziness around the ground truth as humans might also differ quite a lot in how they would write an introduction conditional on a paper's content. Additionally, they could state that test set is rather limited which makes statistical inference difficult, especially given that the effect sizes are quite small sometimes (see above).

**Strengths And Weaknesses:**

Strengths:
- The authors propose an interesting approach of generating an introduction based on the paper's actual content, compared to conditioning on the title/abstract.
- The idea of constructing an explicit reasoning graph increases the interpretability and the grounding in the paper's content.

Weaknesses:
- I might misunderstand, but to me the evaluation seems circular. The training reward uses the ground-truth introduction for entity coverage and paper-consistency terms, and also uses an LLM-as-a-judge for academic quality. The evaluation metrics are then again heavily reference-based and judge-based. This makes it hard to know whether the method learned better scientific reasoning, or mainly learned to optimize proxy metrics tied to the reference Introduction style (overfitting?).
- Additionally, I wonder how much of the benchmark actually measures similarity to human introductions (the ground truth) vs really measuring the quality of the introduction as such. For example, when looking at writing and citation quality, the proposed model is outperformed by OpenAI o3. It actually only underperforms on the similarity and graph-based quality criteria.
- The test set is too small for the strength of the claims. The paper evaluates on only 100 test papers, reports no confidence intervals or significance tests, and then highlights very small differences such as 0.665 vs 0.656 against OpenAI o3.

---

> ### Author Rebuttal · Authors · 2026-03-31
>
> We thank the reviewer for recognizing the task formulation and the reasoning graph's interpretability. We address each concern below.
>
> **W1 & Q1: Evaluation Circularity and Overfitting.** The reviewer's concern centers on whether LECTOR memorizes GT introductions or merely learns to replicate their style and patterns. We argue this is unlikely: the test set is strictly disjoint from the training set, and unlike SFT which provides dense token-level supervision, our RL reward only uses GT as a reference for high-level metrics—the model never receives token-by-token GT signals. As for learning certain writing strategies during RL, we consider this desirable—as long as these preferences improve task performance, they represent genuine capability gains, not overfitting. We provide three lines of evidence:
>
> *(1) Human evaluation.* 8 domain experts scored LECTOR, Base (Qwen3-4B zero-shot), and Original (GT) on 20 test papers across four dimensions (1–5 scale):
>
> | Dimension | Original | Base | LECTOR |
> |---|---|---|---|
> | Logical Coherence | 3.74 | 2.51 | **4.05** |
> | Writing Quality | 3.54 | 2.51 | **3.91** |
> | Citation Integration | **3.21** | 2.06 | 2.99 |
> | Completeness | 3.59 | 2.46 | **3.99** |
> | Overall | 3.52 | 2.38 | **3.73** |
> | Ranked 1st | 25.6% | 0.6% | **73.8%** |
> | Ranked 2nd | 74.4% | 4.4% | 21.2% |
>
> LECTOR was ranked 1st by **73.8%** of 160 expert votes (Spearman ρ=0.815, Krippendorff's α=0.758), breaking the alleged circularity: experts judged quality from their own expertise, not GT similarity.
>
> *(2) SFT baseline rules out GT overfitting.* If the concern is overfitting to GT style, SFT—which imitates GT token-by-token—should perform best. Yet our SFT baseline (same Qwen3-4B) underperforms even zero-shot Base from epoch 1 (WQ: 0.397 vs. 0.546) and degrades further (epoch 5: WQ 0.393). It is SFT that exhibits genuine overfitting, not LECTOR—confirming that LECTOR's gains come from RL-learned reasoning, not GT imitation.
>
> *(3) Qualitative improvements over GT.* In our case study (Appendix C, chiral antiferromagnet AHE paper), LECTOR generates a clear Swales' CARS structure (Territory→Niche→Contribution) with explicit logical flow, while the GT mixes background, materials, and objectives without clear organization. LECTOR explicitly articulates the research gap ("A critical gap in the current understanding lies in...") and states the contribution ("closing a critical gap"), which the GT leaves implicit. These improvements over GT cannot be explained by overfitting.
>
> **W3 & Q2: Statistical Rigor.** We provide comprehensive statistical analyses on all 100 test papers (paired per-paper, LECTOR vs. Base):
>
> | Metric | LECTOR [95% CI] | Base [95% CI] | Δ | Holm p | Cohen's dz |
> |---|---|---|---|---|---|
> | GQ | 0.745 [0.701, 0.784] | 0.478 [0.427, 0.526] | +0.267 | 1.45e-11 | 0.866 |
> | PC | 0.486 [0.474, 0.498] | 0.453 [0.440, 0.467] | +0.032 | 2.07e-08 | 0.675 |
> | WQ | 0.834 [0.814, 0.854] | 0.546 [0.518, 0.573] | +0.289 | 5.89e-16 | 1.592 |
> | CQ | 0.530 [0.500, 0.560] | 0.444 [0.419, 0.468] | +0.086 | 2.91e-06 | 0.517 |
> | OP | 0.665 [0.651, 0.679] | 0.510 [0.498, 0.522] | +0.155 | 5.89e-16 | 1.617 |
>
> All improvements are significant after Holm–Bonferroni correction (all p < 3e-06) with large effect sizes for WQ (dz=1.592) and OP (dz=1.617) and non-overlapping 95% bootstrap CIs (n=10,000). Each of the 100 Nature Communications test papers contains ~6,000 words, making data acquisition costly, yet statistical power is sufficient (all p < 1e-05). Our human evaluation adds 20 papers × 8 experts = 160 independent assessments.
>
> Regarding the LECTOR vs. o3 gap on OP (0.665 vs. 0.656): LECTOR is a 4B open-source model while o3 is a vastly larger commercial model. Comparable performance with orders-of-magnitude fewer parameters is itself significant. The framework can be applied to larger backbones for further gains—the contribution is the methodology, not the model scale.
>
> **W2 & Q3: Similarity vs. Quality and "True" Benchmark.** CCIG is a complex task requiring multi-dimensional metrics (GQ, GW, PC, WQ, CQ). We agree that distinguishing similarity from quality matters. Given that the ultimate goal is introduction writing, **WQ is the most relevant "true" benchmark**, where LECTOR achieves the strongest gain (+52.7% over Base).
>
> On similarity: as noted in W1, without dense token-level supervision, LECTOR does not closely replicate GT text. Our human evaluation confirms this—LECTOR surpasses the Original (GT itself) in 3 of 4 dimensions, and case studies show LECTOR correcting structural weaknesses present in GT. These demonstrate genuine quality improvements beyond GT similarity.
>
> **Limitations.** We acknowledge GT fuzziness—different authors may write introductions differently. This is an inherent limitation of reference-based evaluation, though Nature Communications' rigorous peer review ensures a baseline quality standard.
>
> We hope these address the reviewer's concerns.

---

> > ### Author Rebuttal · Reviewer_tk7z · 2026-04-02
> >
> > Thank you for the rebuttal and adding the human evaluation and paired statistics. However, the new evidence almost entirely targets the LECTOR vs. Base comparison, while my initial concerns were about the LECTOR vs. o3 comparison and the circularity of the evaluation framework. Therefore, the paper’s stronger claims against o3 remain under-supported. Likewise, the SFT ablation does not resolve the benchmark-validity issue: several reward terms and evaluation submetrics remain reference-anchored. Therefore, I keep my initial score.

---

> > > ### Author Response · Authors · 2026-04-07
> > >
> > > We thank the reviewer for the follow-up. We now address the LECTOR vs. o3 comparison and evaluation framework circularity.
> > >
> > > ## Human Evaluation(including o3)
> > >
> > > | Dimension            | Original | o3        | LECTOR    |
> > > | -------------------- | -------- | --------- | --------- |
> > > | Logical Coherence    | 3.74     | 4.00      | **4.05**  |
> > > | Writing Quality      | 3.54     | **4.12**  | 3.91      |
> > > | Citation Integration | 3.21     | **3.51**  | 2.99      |
> > > | Completeness         | 3.59     | 3.21      | **3.99**  |
> > > | Overall              | 3.52     | 3.71      | **3.73**  |
> > > | Ranked 1st           | 10.6%    | **48.8%** | 40.6%     |
> > >
> > > LECTOR and o3 achieve comparable overall scores (3.73 vs. 3.71) with complementary strengths: o3 leads on Writing Quality and Citation Integration, while LECTOR leads on Logical Coherence and Completeness (3.99 vs. 3.21), reflecting the benefit of explicit reasoning graph structure (Spearman ρ = 0.803, Krippendorff's α = 0.758).
> > >
> > > We clarify that we have not claimed LECTOR surpasses o3—our paper states it “achieves parity with GPT-o3”(Lines 361–363) on Overall Performance, and we acknowledge its inferiority on Writing Quality and Citation Quality. Given that LECTOR is a 4B open-source model while o3 is a vastly larger commercial model, comparable overall performance is itself significant. We hope the reviewer will consider that our contribution lies in the methodology, not solely in outperforming a specific model.
> > >
> > > ## Case Studies
> > >
> > > We provide two case studies comparing Original, o3, and LECTOR at [anonymous link](https://anonymous.4open.science/r/anonymous_case_studies-E64D/README.md).
> > >
> > > **LECTOR's strengths.** In both cases, LECTOR produces a clear Swales' CARS structure, while the Originals mix background and contributions in single dense paragraphs. LECTOR explicitly states research gaps (e.g., “a fundamental gap remains in...”), whereas the Originals pose implicit questions. LECTOR also includes more technical depth than o3 (e.g., explicit Bloch Hamiltonian and Chern number formulas in Case 1, XAFS quantitative fitting in Case 2), achieving higher completeness consistent with expert scores.
> > >
> > > **LECTOR's weaknesses.** o3 produces more polished prose and integrates citations more fluently, consistent with its lead on Writing Quality and Citation Integration.
> > >
> > > **Evidence against GT overfitting.** LECTOR differs substantially from the Original in paragraph structure (multi-paragraph CARS vs. single-paragraph), content selection (e.g., Case 2's Original devotes ~500 chars to LaAlO₃/SrTiO₃ interfaces; LECTOR focuses instead on molecular adsorption), gap articulation (declarative statements vs. implicit questions), and technical depth (formulations absent from Original). These differences cannot arise from imitating GT. LECTOR does exhibit a consistent rhetorical style, but as argued in our first rebuttal, learning writing strategies through RL represents genuine capability gains, not overfitting.
> > >
> > > ## Evaluation Framework
> > > Since CCIG is a newly proposed task, it does not yet have the well-established evaluation conventions of mature tasks. We understand the reviewer's caution and address the two aspects of the concern:
> > >
> > > **On reference-anchored metrics.** The reviewer notes that “several reward terms and evaluation submetrics remain reference-anchored.” We clarify that our metrics include both reference-based dimensions (e.g., consistency, entity coverage) and reference-free dimensions (e.g., logical structure, problem clarity). The concern is whether reference-anchored metrics lead the model to imitate GT. We argue this is unlikely for three reasons: (1) train/test sets are strictly disjoint; (2) RL provides only scalar rewards, not token-level supervision; and (3) the case studies confirm LECTOR's outputs differ substantially from GT.
> > >
> > > **On reward–evaluation circularity.** The reviewer's concern may be that using similar metrics for training reward and evaluation creates circularity. We note that this alignment is not a flaw but rather a necessary design choice: our metrics comprehensively assess introduction quality across multiple dimensions, and to our best knowledge, no better alternative exists. Since we consider these metrics valid quality measures, using them to guide RL is the principled choice—unrelated proxy rewards would be counterproductive. This is standard in RL for LLMs: InstructGPT (Ouyang et al., 2022) trains with human-preference rewards and evaluates via human preference; DeepSeek-R1 (Guo et al., 2025) trains with accuracy rewards and evaluates on the same benchmarks. Our setting involves more metrics, which may appear complex, but a comprehensive standard is preferable to an artificially simplified one—there is no reason to adopt a less informative metric in either stage. Aligning reward with evaluation is sound methodology, not circularity, and our human evaluation further serves as independent external validation that breaks any closed loop.

---

### Official Review · Reviewer_Y3Sy · 2026-03-13

**Soundness:** 3
**Presentation:** 3
**Significance:** 2
**Originality:** 3
**Overall Recommendation:** 4
**Confidence:** 3

**Summary:**

This paper introduces Content-Conditional Introduction Generation (CCIG), a new task for generating research paper introductions from the paper’s methodology, results, analyses, and citation list rather than from a generic prompt. The authors argue that introduction writing is fundamentally a reasoning and structuring task, not just a text-generation task, because it must faithfully reflect the paper’s motivation, logic, and contributions while maintaining citation accuracy. To address this, they propose LECTOR, a framework that first builds a Reasoning Logic Graph as an explicit logical blueprint and then uses logic-expression co-reinforcement learning to optimize both reasoning fidelity and writing quality. On a large dataset of 10,200 papers, LECTOR improves graph quality, citation quality, and paper consistency, and a 4B model reportedly reaches performance comparable to strong commercial systems.

**Compliance With Llm Reviewing Policy:**

Affirmed.

**Final Justification:**

Rebuttal solved most of my concern

**Key Questions For Authors:**

Please refer to weaknesses

**Limitations:**

yes

**Strengths And Weaknesses:**

Strengths:

1.The paper clearly explains why introduction writing should be treated as a structured reasoning problem rather than ordinary generation. Framing it as CCIG is a meaningful conceptual contribution.

2.The use of a Reasoning Logic Graph is a notable strength because it makes the generation process more explicit and verifiable, instead of relying purely on opaque prompting.

3.The paper does more than propose a method: it also introduces evaluation dimensions for logic fidelity, fluency, and citation quality, and validates the approach on a large real-paper dataset, which makes the claims more convincing.

Weaknesses:

1.it is unclear whether the quality of the Introduction sections or the derived supervision signals was verified by human annotators. Since the method relies heavily on the assumption that the collected Introductions faithfully reflect strong logical structure and citation quality, the absence of human quality control raises concerns about noise in the training and evaluation data.

2.the paper does not fully justify whether this explicit intermediate representation is truly necessary for introduction writing. Human authors generally do not construct formal logic graphs before writing introductions, so it remains unclear whether the graph is an essential ingredient for better performance or simply an added modeling constraint. Stronger ablations or alternative structured planning baselines would help clarify this point.

3.it would be valuable to compare against stronger inference-time scaling approaches, such as multi-pass planning, self-refinement, or test-time reasoning strategies. This is especially important because the task is framed as one of reasoning and structuring, where inference-time scaling could plausibly yield substantial gains without requiring a new training framework.

4.a significant portion of the evaluation seems to depend on LLM-based judgment for dimensions such as logic fidelity or overall writing quality. While this is understandable for a complex generation task, it makes it difficult to quantitatively establish the true performance gap between methods. Without stronger human evaluation or external objective validation, the reported improvements may be sensitive to judge bias or prompt design.

---

> ### Author Rebuttal · Authors · 2026-03-31
>
> We thank the reviewer for recognizing CCIG, the Reasoning Logic Graph, and our evaluation design. We address each concern below.
>
> **W4: LLM-as-Judge Reliability.** We conducted a human evaluation with 8 domain experts on 20 randomly sampled test papers. Each expert scored LECTOR, Base (Qwen3-4B zero-shot), and Original (GT) across four dimensions (1–5 scale) and ranked the three introductions:
>
> | Dimension | Original | Base | LECTOR |
> |---|---|---|---|
> | Logical Coherence | 3.74 | 2.51 | **4.05** |
> | Writing Quality | 3.54 | 2.51 | **3.91** |
> | Citation Integration | **3.21** | 2.06 | 2.99 |
> | Completeness | 3.59 | 2.46 | **3.99** |
> | Overall | 3.52 | 2.38 | **3.73** |
> | Ranked 1st | 25.6% | 0.6% | **73.8%** |
> | Ranked 2nd | 74.4% | 4.4% | 21.2% |
>
> Human experts exhibited evaluation patterns closely aligned with our LLM-as-Judge. In the ranking task, LECTOR was ranked 1st by **73.8%** of 160 expert votes (118/160), while Base was ranked last in 95.0% of cases. The Spearman correlation between human and LLM judgments is ρ=**0.815** (p<0.001), with Krippendorff's α=**0.758**, confirming substantial agreement. These results directly validate the reliability of our automatic evaluation framework.
>
> **W1: Training/Evaluation Data Quality.** All papers are from *Nature Communications*, a Q1 peer-reviewed journal ensuring high-quality introductions. Moreover, we specifically selected papers in the Physics domain, which demand rigorous logical structure inherent to the natural sciences. Our human evaluation indirectly validates this: GT introductions scored 3.52/5 and were ranked 2nd in 74.4% of votes, confirming they serve as reliable supervision signals.
>
> **W2: Necessity of the Logic Graph.** We address this from two perspectives:
>
> *(1) Conceptual justification.* Human authors do not draw formal logic graphs, but they implicitly plan the logical flow before writing—deciding what to cover, how to transition between topics, and how to build toward the final conclusions. The Reasoning Logic Graph explicitly formalizes this implicit planning process, analogous to Chain-of-Thought prompting: humans do not write explicit reasoning chains, yet CoT dramatically improves LLM reasoning. Similarly, the Logic Graph improves generation by making the reasoning structure learnable and verifiable.
>
> *(2) Experimental evidence.* Table 2 in the paper shows that removing the graph (One-Step baseline) degrades PC from 0.486 to 0.476, WQ from 0.834 to 0.829, and CQ from 0.530 to 0.477. Furthermore, Table 3 demonstrates that graph quality has a significant impact on final writing quality—higher GQ scores consistently lead to better WQ, PC, and CQ outcomes—further confirming the functional role of the Logic Graph as a meaningful intermediate representation rather than a redundant constraint.
>
> **W3: Comparison with Inference-Time Scaling.** We appreciate this insightful suggestion. We would like to clarify that LECTOR is not intended as a general-purpose LLM reasoning method, but rather a task-specific framework designed for the structured reasoning demands of CCIG. Inference-time scaling strategies (e.g., Best-of-N, self-refinement, multi-pass planning) are not in conflict with our training-time approach—they are complementary techniques that could be applied on top of LECTOR to further boost performance, rather than competing alternatives. We argue that training-time and inference-time scaling serve different purposes: LECTOR's RL training *internalizes* logical reasoning into a compact 4B model, whereas inference-time methods trade compute for quality at each generation. In Table 1, our 4B LECTOR already matches GPT-o3 in overall performance (OP: 0.665 vs 0.656), demonstrating the efficiency of training-time optimization. We plan to explore combining LECTOR with inference-time scaling in future work and thank the reviewer for this suggestion.
>
> **Summary.** We have provided substantial new evidence and detailed explanations to address each concern: (1) human evaluation with 8 domain experts showing ρ=0.815 human–LLM agreement and 73.8% first-rank preference for LECTOR, directly validating our automatic evaluation; (2) data quality assurance through Nature Communications' peer-review process and Physics-domain selection, further confirmed by GT introductions scoring 3.52/5 in human evaluation; (3) ablation and baseline experiments (Tables 2–3) confirming the functional necessity of the Logic Graph; and (4) a clarification that training-time and inference-time scaling are complementary, with plans to explore their combination in future work. We hope these results and explanations address the reviewer's concerns and welcome further discussion.

---

> > ### Author Rebuttal · Reviewer_Y3Sy · 2026-04-03
> >
> > Thank you for the new results and analysis. I will raise my score

---

> > > ### Author Response · Authors · 2026-04-07
> > >
> > > We are glad that our rebuttal has addressed the concerns, and thank the reviewer for the constructive questions and suggestions throughout the review process. The recognition of our work and the updated score are greatly appreciated.

---

### Decision · Program_Chairs · 2026-04-30

**Decision:**

Accept (regular)

**Comment:**

This paper formulates the Content-Conditional Introduction Generation (CCIG) task and proposes LECTOR, a Logic-Expression Co-Reinforcement Learning framework that constructs a reasoning logic graph from a paper's main body and then jointly optimizes the graph's structural fidelity and the generated narrative's quality via RL. The core contribution lies in reframing scientific introduction writing as a structured reasoning and grounding problem rather than a pure text-generation problem, supported by a new dataset of 10,200 Nature Communications papers and a multi-dimensional evaluation protocol.

Strengths and weaknesses raised by reviewers:

- **Strengths:**
  - The CCIG formulation is meaningful and more realistic than title/abstract-conditioned generation, treating introduction writing as a structured reasoning task (Y3Sy, tk7z, 2Wn5, iWNk).
  - The use of an explicit Reasoning Logic Graph as an interpretable intermediate representation/information bottleneck helps ground generation in the paper's content and improves verifiability (Y3Sy, tk7z, 2Wn5, iWNk).
  - Solid empirical setup with a large real-paper dataset, multi-dimensional evaluation metrics, and useful ablations that demonstrate the contributions of graph quality and joint optimization (Y3Sy, iWNk); the memory-efficient RL pipeline without a separate reward model is a practical plus (2Wn5).

- **Weaknesses:**
  - Potential evaluation circularity and overfitting risk: both training rewards and evaluation metrics are heavily anchored to the ground-truth introduction, making it hard to disentangle genuine reasoning gains from reference-style imitation (tk7z, iWNk).
  - Heavy reliance on LLM-as-a-judge (Qwen3-235B) for core quality dimensions, with limited initial human evaluation and concerns about judge bias/reward hacking (Y3Sy, 2Wn5, iWNk).
  - Concerns about the Graph-Write Alignment drop after RL training, which reviewers feared could indicate the model drifting from the graph to chase fluency (2Wn5), as well as a weak baseline setup lacking an SFT comparison in the original submission (2Wn5).
  - Limited statistical rigor and small test set (100 papers) relative to the strength of claims, with especially narrow margins against o3 on OP (0.665 vs. 0.656) that remained under-supported against the strongest commercial baseline (tk7z).

The reviewer consensus after rebuttal is positive: Y3Sy and iWNk indicate their concerns are fully resolved and raised their scores, 2Wn5 is partially resolved and also raised the score, and only tk7z maintains a reject citing unresolved circularity and LECTOR-vs-o3 concerns. Given that the added human evaluation (ρ=0.815 with LLM judges, 73.8% first-rank preference), SFT baseline, and paired statistical tests addressed the majority of concerns, and the task formulation plus logic-graph methodology are a contribution others are likely to build on, the paper meets the bar for acceptance despite remaining limitations around reference-anchored evaluation.